# Reactive astrocytes function as phagocytes after brain ischemia via ABCA1-mediated pathway

Yosuke M. Morizawa[1,2], Yuri Hirayama[1], Nobuhiko Ohno[3], Shinsuke Shibata[4], Eiji Shigetomi[1], Yang Sui[3], Junichi Nabekura[5,6], Koichi Sato[7], Fumikazu Okajima[7], Hirohide Takebayashi[8], Hideyuki Okano[4] & Schuichi Koizumi [1]

Astrocytes become reactive following various brain insults; however, the functions of reactive astrocytes are poorly understood. Here, we show that reactive astrocytes function as phagocytes after transient ischemic injury and appear in a limited spatiotemporal pattern. Following transient brain ischemia, phagocytic astrocytes are observed within the ischemic penumbra region during the later stage of ischemia. However, phagocytic microglia are mainly observed within the ischemic core region during the earlier stage of ischemia. Phagocytic astrocytes upregulate ABCA1 and its pathway molecules, MEGF10 and GULP1, which are required for phagocytosis, and upregulation of ABCA1 alone is sufficient for enhancement of phagocytosis in vitro. Disrupting ABCA1 in reactive astrocytes result in fewer phagocytic inclusions after ischemia. Together, these findings suggest that astrocytes are transformed into a phagocytic phenotype as a result of increase in ABCA1 and its pathway molecules and contribute to remodeling of damaged tissues and penumbra networks.

[1] Department of Neuropharmacology, Interdisciplinary Graduate School of Medicine, University of Yamanashi, Chuo, Yamanashi 409-3898, Japan. [2] Department of Super-network Brain Physiology, Graduate School of Life Science, Tohoku University, Sendai, Miyagi 980-8575, Japan. [3] Division of Neurobiology and Bioinformatics, National Institute for Physiological Sciences, Okazaki, Aichi 444-8585, Japan. [4] Department of Physiology and Electron Microscope Laboratory, Keio University School of Medicine, Shinjuku, Tokyo 160-8582, Japan. [5] Division of Homeostatic Development, National Institute for Physiological Sciences, Okazaki, Aichi 444-8585, Japan. [6] Department of Physiological Sciences, The Graduate School for Advanced Study, Hayama, Kanagawa 240-0193, Japan. [7] Laboratory of Signal Transduction, Institute for Molecular and Cellular Regulation, Gunma University, Maebashi, Gunma 371-8512, Japan. [8] Division of Neurobiology and Anatomy, Graduate School of Medical and Dental Sciences, Niigata University, Niigata, Niigata 951-8510, Japan. Yuri Hirayama and Nobuhiko Ohno contributed equally to this work. Correspondence and requests for materials should be addressed to S.K. (email: skoizumi@yamanashi.ac.jp)

Brain ischemia is one of the leading causes of death and chronic adult disability in humans and results from an interrupted blood supply to the brain, resulting in cell death[1]. Astrocytes are highly responsive resident brain cells that dramatically change their characteristic to brain damage and are thus termed "reactive astrocytes"[2, 3]. Previous reports showed reactive astrocytes release trophic factors, synaptogenic factors and extracellular matrix, which promote neuronal survival, synapse formation and plasticity, indicating astrocytes participate in remodeling of the central nervous system after ischemia[1–7].

After brain damage occurs, neuronal circuits and the local environment are disrupted causing the collection of debris in the affected region. The rapid engulfment and clearance of such dead cells or debris is essential for the remodeling of the neuronal circuits and/or microenvironment[8–10]. So far, the engulfment has been thought to be limited to professional phagocytes, i.e., microglia in the brain[11, 12]. However, here is growing evidence that non-professional phagocytes can also participate in that process[8, 13]. Previous studies have shown the presence of degenerated axons and apoptotic neurons in astrocytes in injured

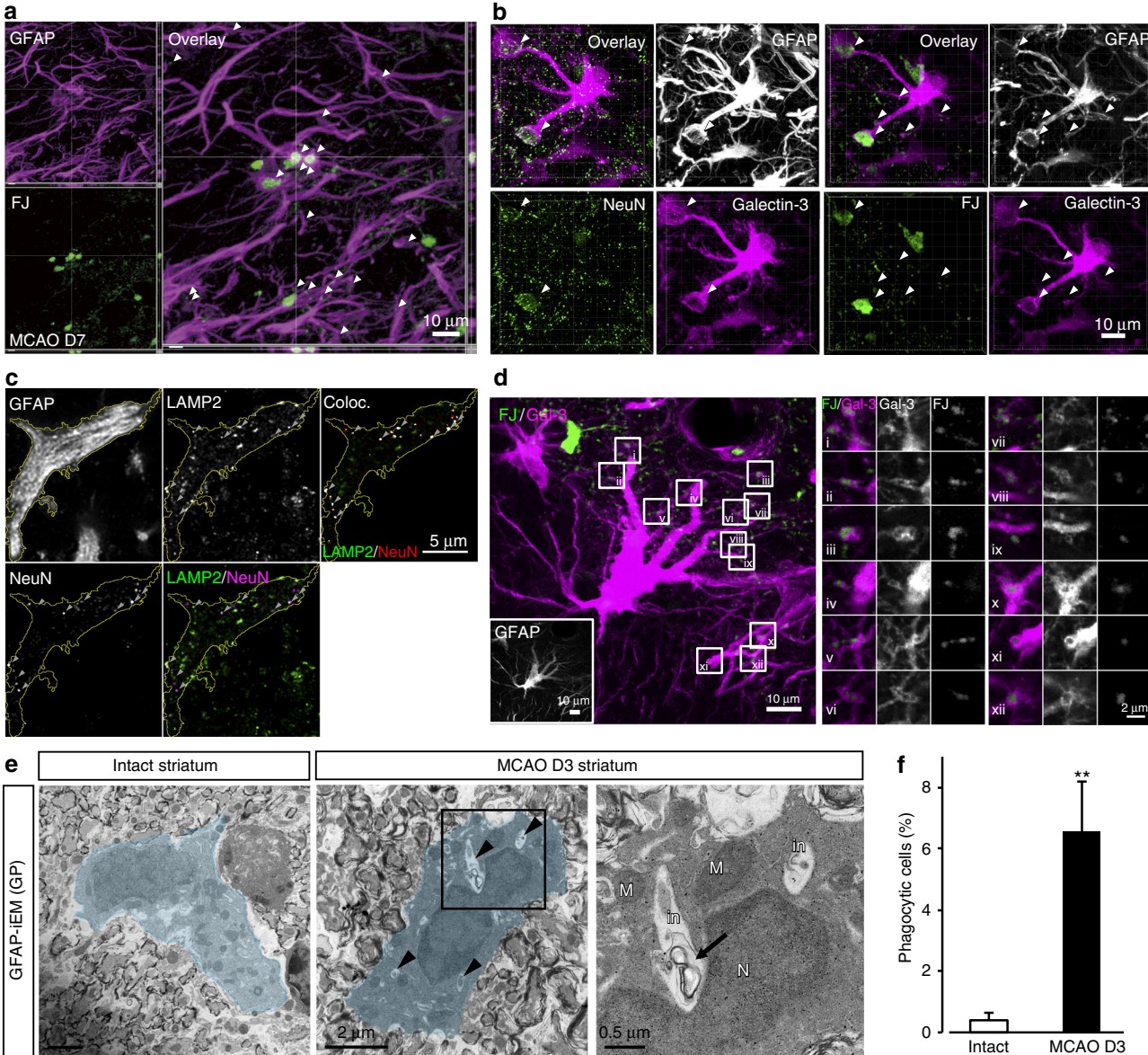

**Fig. 1** Astrocytes become phagocytic after transient ischemic injury in vivo. **a** Representative images of the ischemic penumbra showing FJ-labeled and GFAP-immunostained brain sections at 7 days after MCAO (n = 10 mice). *Arrowheads* indicate FJ+-degenerating neuronal debris enwrapped by GFAP+ astrocytes. Fourteen images per z-stack image (0.49 μm step). **b** Representative image of GFAP+, Galectin-3+ astrocyte enwrapped NeuN+, FJ+ large neuronal debris. *Arrowheads* indicate FJ+ degenerating neuronal debris enwrapped by GFAP+ astrocytes. Thirty-four images per z-stack image (0.3 μm step). **c** Representative images of NeuN+ signals in LAMP2+ lysosomes in GFAP+ astrocytes (*arrowheads*). **d** Processes of Galectin-3 and GFAP double-positive astrocytes enwrapping numerous FJ+ small debris (n = 8 mice). The *white squares* in the *left panel image* indicate the region of high magnification of shown in the right panel. **e** Immunoelectron microscopic (iEM) images of GFAP+ astrocytes (gold particles: GP; *blue*). *Left*, an image of an astrocyte in the striatum of an intact mouse. *Middle*, an image of an astrocyte in the ipsilateral striatum at 3 days after MCAO. *Arrowheads* indicate phagocytic inclusions. *Right*, high-magnification image of the box shown in the *middle panel*. *Arrow* indicates myelin-like structure. N, nucleus; M, mitochondria; in, phagocytic inclusion. **f** Percentage of GFAP+ astrocytes with phagocytic inclusions in total in the intact (n = 461 cells, 4 mice) and MCAO-treated striatum (n = 315 cells, 4 mice, **P = 0.0056, unpaired t-test). Values represent means ± SEM

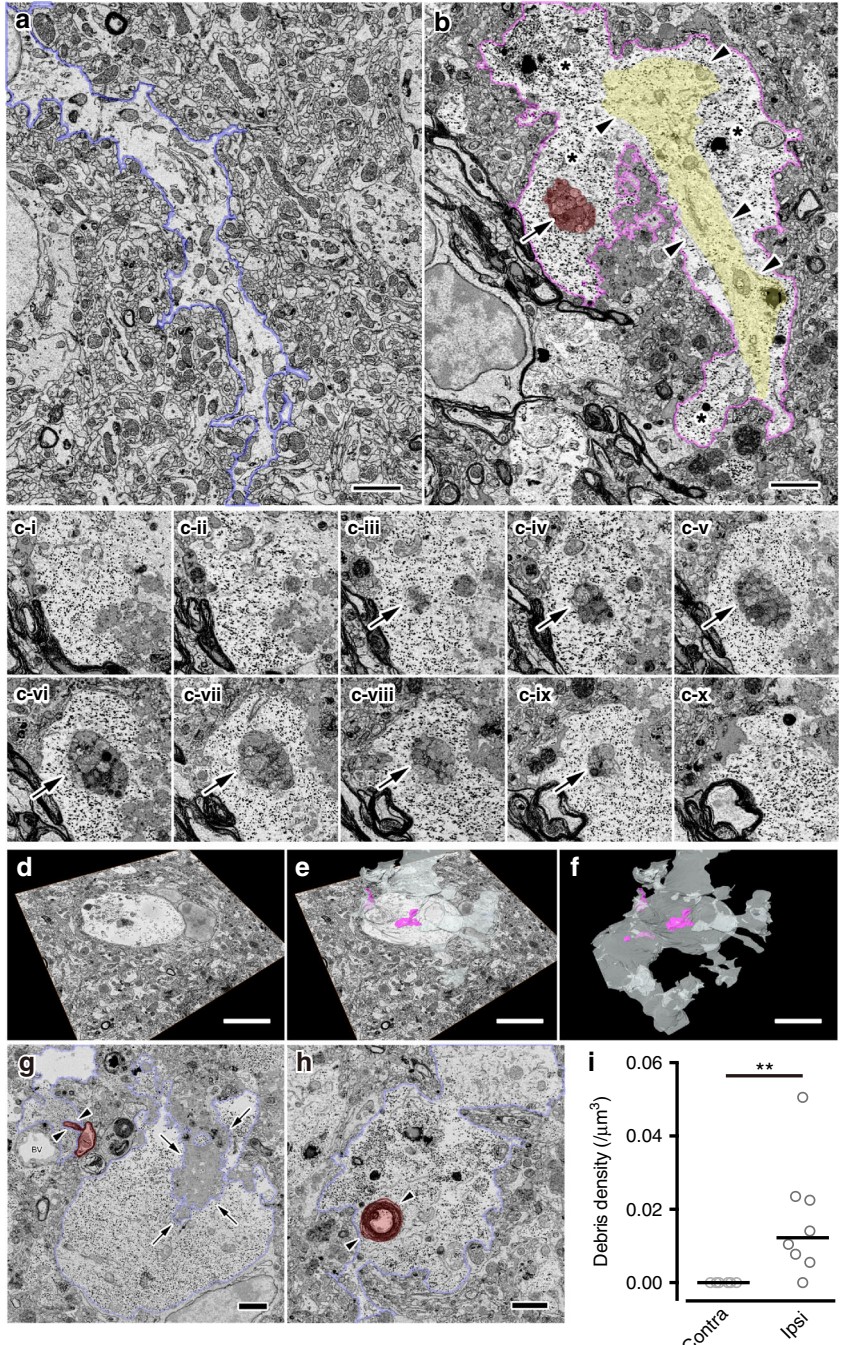

**Fig. 2** Phagocytosis of cellular debris by reactive astrocytes in the ischemic penumbra. **a–c** Compared with astrocyte processes in the contralateral striatum (**a**, *blue*), those in the ischemic penumbra were characterized by prominent intermediate filament bundles (**b**, *yellow, arrowheads*) and abundant glycogen granules (**b**, *asterisks*), and often contained small pieces of cellular debris (**b**, *red, arrow*). The debris was completely included in the astrocytic cytoplasm (c-i–c-x, *arrows*). **d–f** Three-dimensional reconstruction shows several pieces of debris (**e**, **f**, *pink*) are scattered within single processes (**e**, **f**, *white*). **g, h** Astrocyte processes surround blood vessels (**g**, BV), and engulf large debris (**g**, *arrows*) and myelin debris (**g**, **h**, *red, arrowheads*). Bars: 2 μm **a**, **b**, **g**, **h** or 5 μm **d–f**. **i** Dot plots of debris density (number/volume (μm³)) in astrocytes in the contralateral and ipsilateral striatum ($n = 5$, eight cells, three mice. $**P < 0.01$ vs. control, Mann–Whitney $U$-test)

brains[14–16]. Additionally, recent studies have shown that optic nerve head astrocytes constitutively engulf axonal materials, even under normal physiological conditions[17, 18]. A gene profiling study suggested that astrocytes are enriched in genes involved in engulfment pathways, including phagocytic receptors, intracellular molecules, and opsonins, in the developing mouse forebrain[19], and a recent study revealed that immature astrocytes actively participate in synapse elimination in the developing

retinogeniculate system[20]. Although accumulating evidence suggests that astrocytes may also participate in clearance in the brain, astrocytic phagocytosis received limited attention and the mechanisms, physiological consequences and difference from microglia remain poorly understood.

The present study showed that a subset of reactive astrocytes within the ischemic penumbra region is transformed into phagocytic cells following transient ischemic injury in the adult

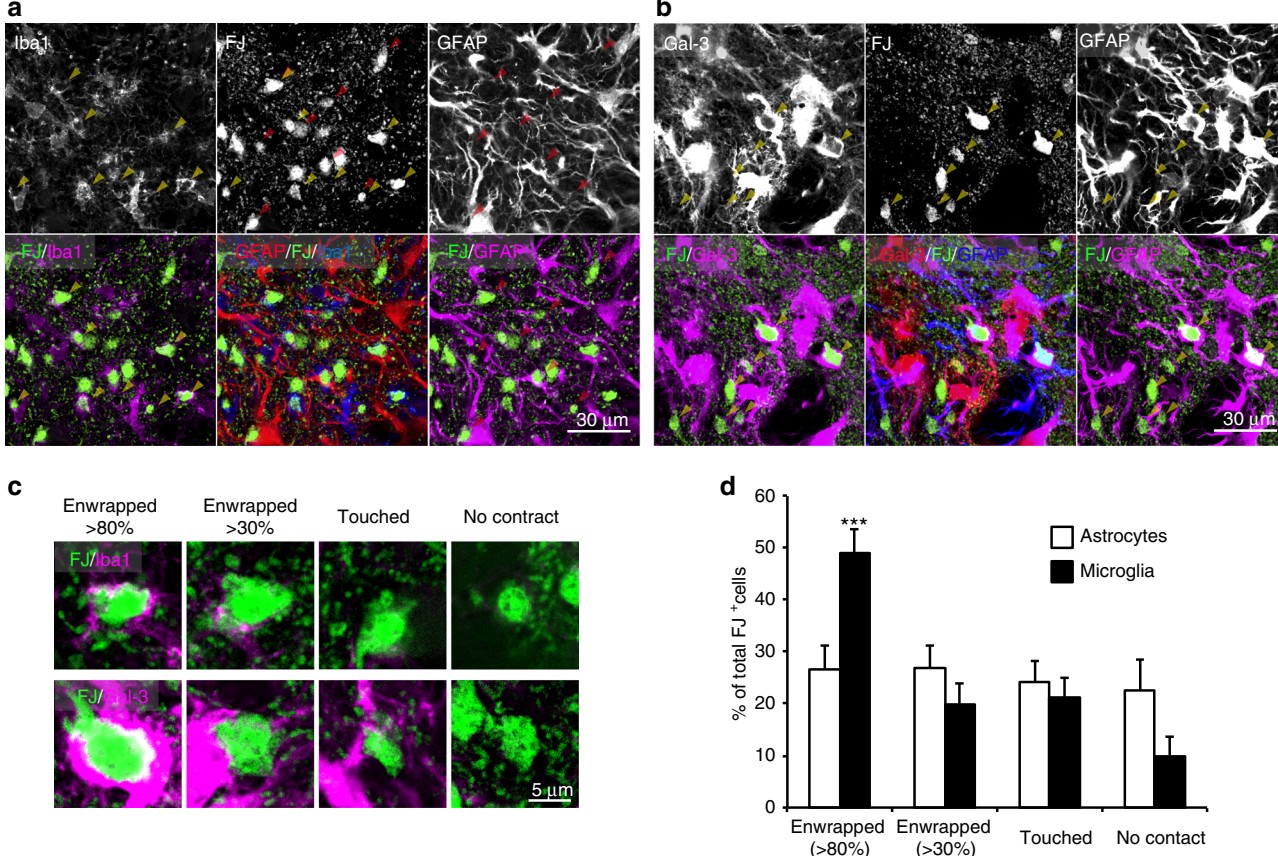

**Fig. 3** Comparison of enwrapped neuronal debris by astrocytes and microglia. **a** Representative images showing immunohistochemical (IHC) staining for GFAP, and Iba1 with FJ staining in the ischemic penumbra at 7 days after MCAO. Iba1+ microglia (*yellow arrowheads*) and GFAP+ astrocytes (*red arrowheads*) enwrapped FJ+ large neuronal debris by their processes. **b** Representative images showing IHC staining for Galectin-3 and GFAP with FJ staining in the ischemic penumbra at 7 days after MCAO. Galectin-3 and GFAP double-positive processes enwrapped FJ+ large neuronal debris (yellow arrowheads). **c** Representative images showing quantification of the amount of FJ+ large debris (size over 10 μm$^2$) enwrapped by Iba1+ microglia or Galectin-3+, GFAP+ astrocytes. **d** Quantification of FJ+ large debris enwrapment by Iba1+ microglia and Galectin-3+, GFAP+ astrocytes ($n = 13$ fields, four mice, ***$P < 0.001$ vs. astrocytes). All images are single plane images

brain. We identified ATP-binding cassette transporter A1 (ABCA1) and molecules in its pathway, multiple EGF-like-domains 10 (MEGF10) and engulfment adapter phosphotyrosine-binding domain containing 1 (GULP1), as the responsible molecules for astrocytic phagocytosis. We also report that astrocytic phagocytosis displayed distinct spatiotemporal pattern from microglial ones. Together these findings suggest that astrocytes can become phagocytic in the pathological brain and contribute to clearance or brain remodeling in the penumbra region, with characteristics different from microglia.

## Results

**Reactive astrocytes show phagocytic features after ischemia.** Brain injury leads to the accumulation of substantial amounts of neural waste in the damaged core, as well as in the non-damaged peri-infarct region (hereinafter called penumbra), where astrocytes become reactive. To determine whether reactive astrocytes become phagocytic under pathological conditions, we employed a transient middle cerebral artery occlusion (MCAO) mouse model[21, 22]. The mice were subjected to right-sided ligature MCAO for 15 min followed by various periods of reperfusion. We initially assessed the MCAO-evoked neuronal damage using a specific marker for neuronal degeneration, Fluoro-jade B (FJ)[23, 24]. FJ-positive (FJ+) signals, i.e., degenerating neurons

and debris including dendrites, axons, and nerve terminals, were observed in the ipsilateral striatum (Supplementary Fig. 1a). We confirmed FJ+ large somatic signals were entirely colocalized with weak NeuN+ neurons, which correlated with reduced MAP2+ signals (Supplementary Fig. 1c, d). As expected, strong GFAP+ signals were found mainly in the penumbra region surrounding the ischemic core, where Iba1+ microglia were mainly found. Both GFAP+ astrocytes and Iba1+ microglia (including macro-phages or other immune cells) were transformed into a "reactive state" with hypertrophic somata and thickening of processes (Supplementary Fig. 1b). Surprisingly, FJ and NeuN-double positive degenerating neurons and small neuronal debris were enclosed by GFAP+ astrocytes in the ischemic penumbra 7 days after MCAO (Fig. 1a, b). Additionally, we immunostained penumbra astrocytes for the lysosome marker LAMP2 to confirm whether they contained machinery to digest engulfed debris. LAMP2+ signals colocalized with NeuN+ signals in reactive astrocytes (Fig. 1c), indicating that a potential role for reactive astrocytes as phagocytes. Some NeuN+ signals did not colocalize with LAMP2+ signals in astrocytes suggesting that these NeuN+ signals should be in phagosomal compartments prior to lysoso-mal degradation.

Galectin-3, a phagocytic biomarker[17, 25] was found in reactive astrocytes, which spatially correlated with FJ+ signals. Galectin-3+ astrocytes enwrapped FJ+ large signals and a number of FJ+ small

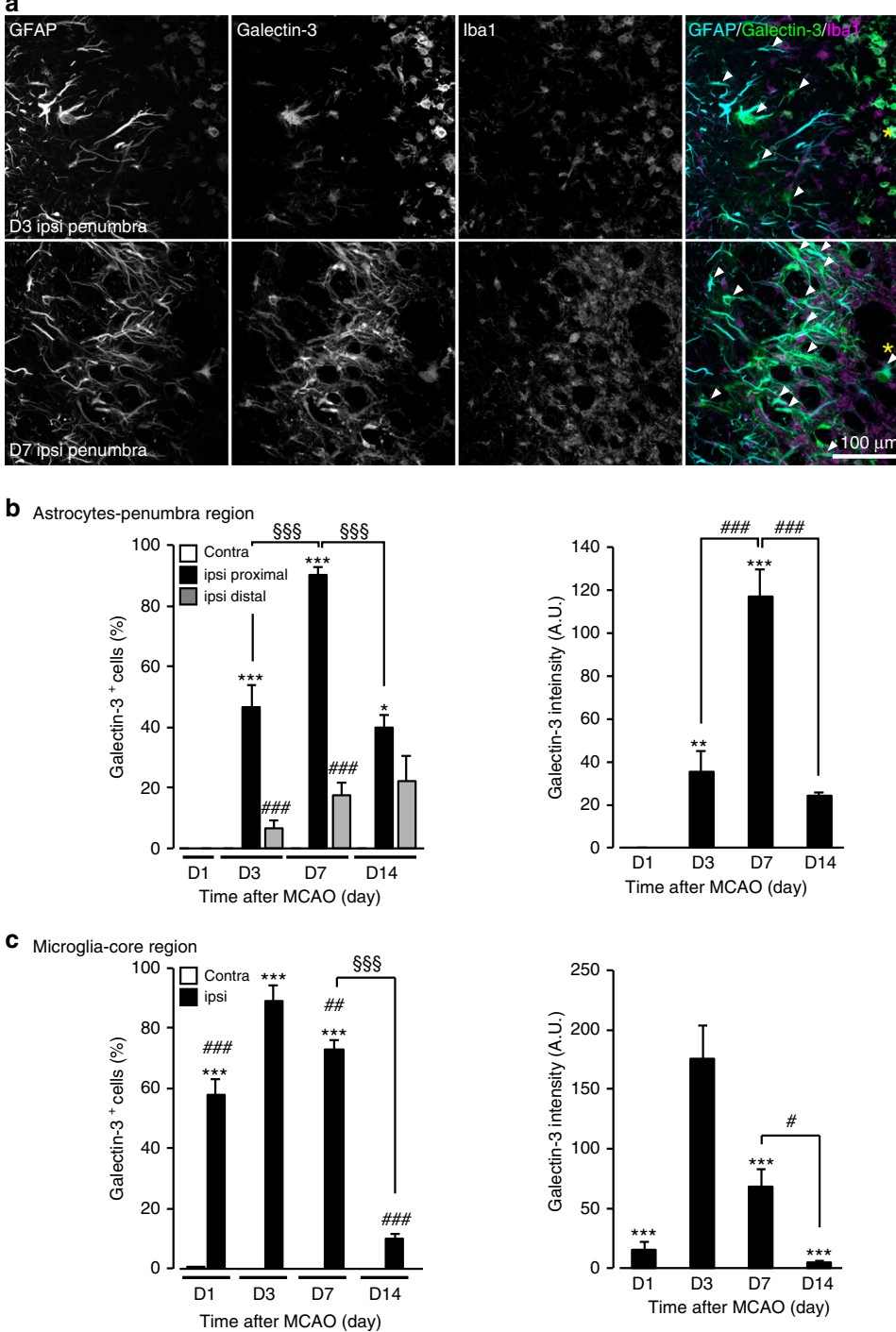

**Fig. 4** Spatiotemporal differences in Galectin-3 between astrocytes and microglia. **a** Representative images showing IHC staining for GFAP, Iba1, and Galectin-3 in the ischemic penumbra at 3 and 7 days after MCAO. *Arrowheads* indicate Galectin-3 in astrocytes. *Asterisks* indicate the ischemic core ($n = 5$ mice). Thirteen images per z-stack image (0.5 μm step). **b** *Left*, quantification of spatiotemporal profile of Galectin-3+ astrocytic population (3PGDH+ cells) after MCAO ($n = 4$, 4, 5, 15, 15, 15, 4, 12, 12, 4, 12, 12 fields, 3–5 mice, *$P < 0.05$, ***$P < 0.001$ vs. contra (corresponding day), ###$P < 0.001$ ipsi proximal vs. ipsi distal, §§§$P < 0.001$ vs. ipsi proxymal D7, one-way ANOVA ($P < 0.0001$) with Tukey's multiple comparison test). *Right*, quantification of spatiotemporal profile of Galectin-3 immunoreactivity mean intensity in 3PGDH+ astrocytes in the ipsilateral striatum after MCAO ($n = 30$, 51, 208, 76 cells, 3–5 mice, **$P < 0.01$, ***$P < 0.001$ vs. D1, ###$P < 0.001$ vs. D7, one-way ANOVA ($P < 0.0001$) with Tukey's multiple comparison test). **c** *Left*, quantification of mean intensity of Galectin-3 immunoreactivity in Iba1+ microglia after MCAO ($n = 12$, 12, 15, 15, 10, 14, 10, 15 fields, 4–5 mice, ***$P < 0.001$ vs. contra (corresponding day), ##$P < 0.01$, ###$P < 0.001$ vs. ipsi D3, §§§$P < 0.001$ vs. ipsi D7, one-way ANOVA ($P < 0.0001$) with Tukey's multiple comparison test). *Right*, quantification of Galectin-3+ microglia population (Iba1+) in the ipsilateral striatum after MCAO ($n = 12$, 15, 14, 15 fields, 4–5 mice, ***$P < 0.001$ vs. D3, #$P < 0.05$ vs. D7, one-way ANOVA ($P < 0.0001$) with Tukey's multiple comparison test). Values represent means ± SEM

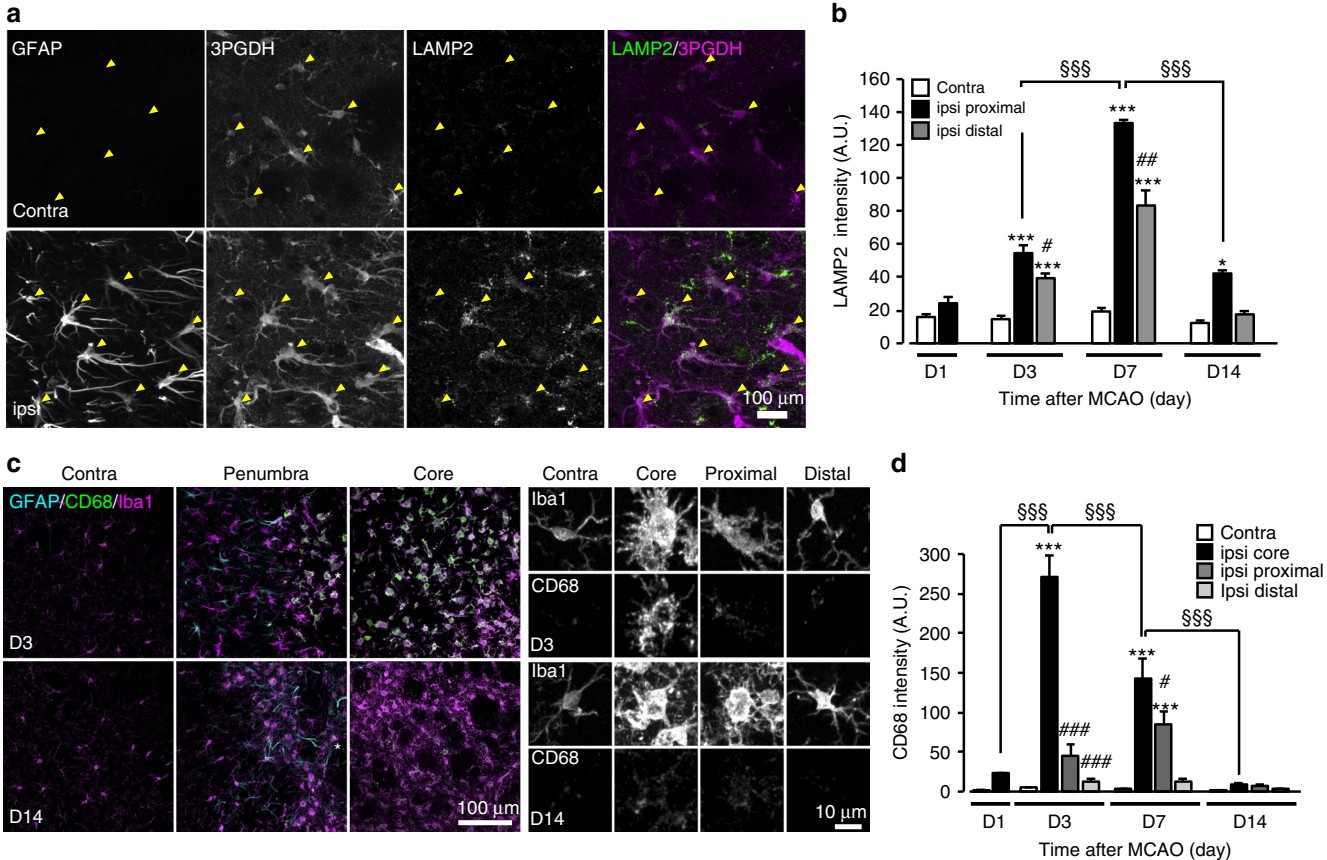

**Fig. 5** Spatiotemporal differences in lysosomal protein between astrocytes and microglia. **a** Representative images of GFAP, 3PGDH, and LAMP2 immunoreactivity in the contralateral striatum (*upper*) and the ischemic penumbra of the ipsilateral striatum (*lower*) at 7 days after MCAO. *Arrowheads* indicate astrocytes (n = 10 mice). Forty images per z-stack image (0.47 μm step). **b** Quantification of LAMP2 immunoreactivity mean intensity in 3PGDH$^+$ astrocytes after MCAO (n = 9, 9, 9, 16, 24, 9, 18, 26, 9, 16, 24 fields, 3, 4 mice, *P < 0.05, ***P < 0.001 vs. contra (corresponding day), $^\#$P < 0.05, $^{\#\#}$P < 0.01 ipsi (proximal) vs. ipsi (distal), $^{\S\S\S}$P < 0.001 vs. ipsi proximal D7, one-way ANOVA (P < 0.0001) with Tukey's multiple comparison test). **c** Representative images of GFAP, CD68, and Iba1 immunoreactivity in the ischemic core, penumbra, and contralateral striatum at 3 and 14 days after MCAO (n = 4 mice). High-magnification images are shown in the *right panel*. Eighteen images per z-stack image (1.0 μm step). **d** Quantification of CD68 immunoreactivity mean intensity in Iba1$^+$ microglia after MCAO (n = 8, 8, 10, 8, 4, 4, 12, 11, 10, 10, 8, 8, 8, 8 fields, 4–6 mice, ***P < 0.001 vs. contra (corresponding day), $^\#$P < 0.05, $^{\#\#\#}$P < 0.001 vs. ipsi core (corresponding day), $^{\S\S\S}$P < 0.001 vs. respective ipsi core, one-way ANOVA (P < 0.0001) with Tukey's multiple comparison test). Values represent means ± SEM

fractions, suggesting that a subset of reactive astrocytes within the ischemic penumbra changed into phagocytic cells following ischemic injury (Fig. 1a, b, d).

Immunoelectron microscopic (iEM) analysis was performed to confirm astrocytic phagocytosis of debris. As shown in Fig. 1e, immunolabeled glial fibrillary acidic protein (GFAP)-positive astrocytes revealed a typical morphology with many mitochondria, a clear cytoplasm and some processes in the intact striatum, but did not exhibit phagocytic inclusions (Fig. 1e). By 3 days after MCAO, the reactive astrocytes exhibited many phagocytic inclusions within their cytoplasm (Fig. 1e). These phagocytic astrocytes were frequently detected in the ipsilateral striatum, which is summarized in Fig. 1f. Electron microscopic (EM) analysis showed that reactive astrocytes were capable of phagocytizing multiple components because they engulfed myelin-like structure (Figs. 1e and 2h), synaptophysin1$^+$ synapses and unidentified components (Supplementary Fig. 2). We also found that Iba1$^+$ signals localized with LAMP2$^+$ lysosomal vesicles in the reactive astrocytes, indicating that phagocytic astrocytes could engulf immune cell debris (Supplementary Fig. 3).

To confirm that reactive astrocytes engulfed debris completely, the ischemic penumbra was also imaged with serial block face scanning electron microscopy (SBF-SEM; Supplementary Fig. 4). Consistent with the observation of immunohistochemistry (IHC) and iEM experiments, reactive astrocytes often contained small pieces of cellular debris, which were located specifically in their cytoplasm (Fig. 2). In contrast, astrocytes in the contralateral striatum contained no cellular debris (Fig. 2i, 7/8 astrocytes had phagocytic inclusions in the ipsilateral striatum, and no phagocytic inclusions were observed in five astrocytes in the contralateral striatum 7 days after MCAO), indicating that reactive astrocytes acquired phagocytic function after brain ischemia. Interestingly, our 3D-EM analysis revealed that large debris was enwrapped by astrocytes and appeared to be in the process of being engulfed by astrocytic processes (Fig. 2g).

**Enwrapment of neuronal debris by astrocytes and microglia.** Microglia mediate damaged cell and debris clearance after acute brain damage. To investigate how frequently astrocytes engulf debris compared with microglia, we measured the enwrapment of FJ$^+$ large neuronal debris by astrocytes and microglia as an index of engulfment. We compared IHC staining for Iba1, GFAP and Galectin-3 with FJ staining, and analyzed the rim of

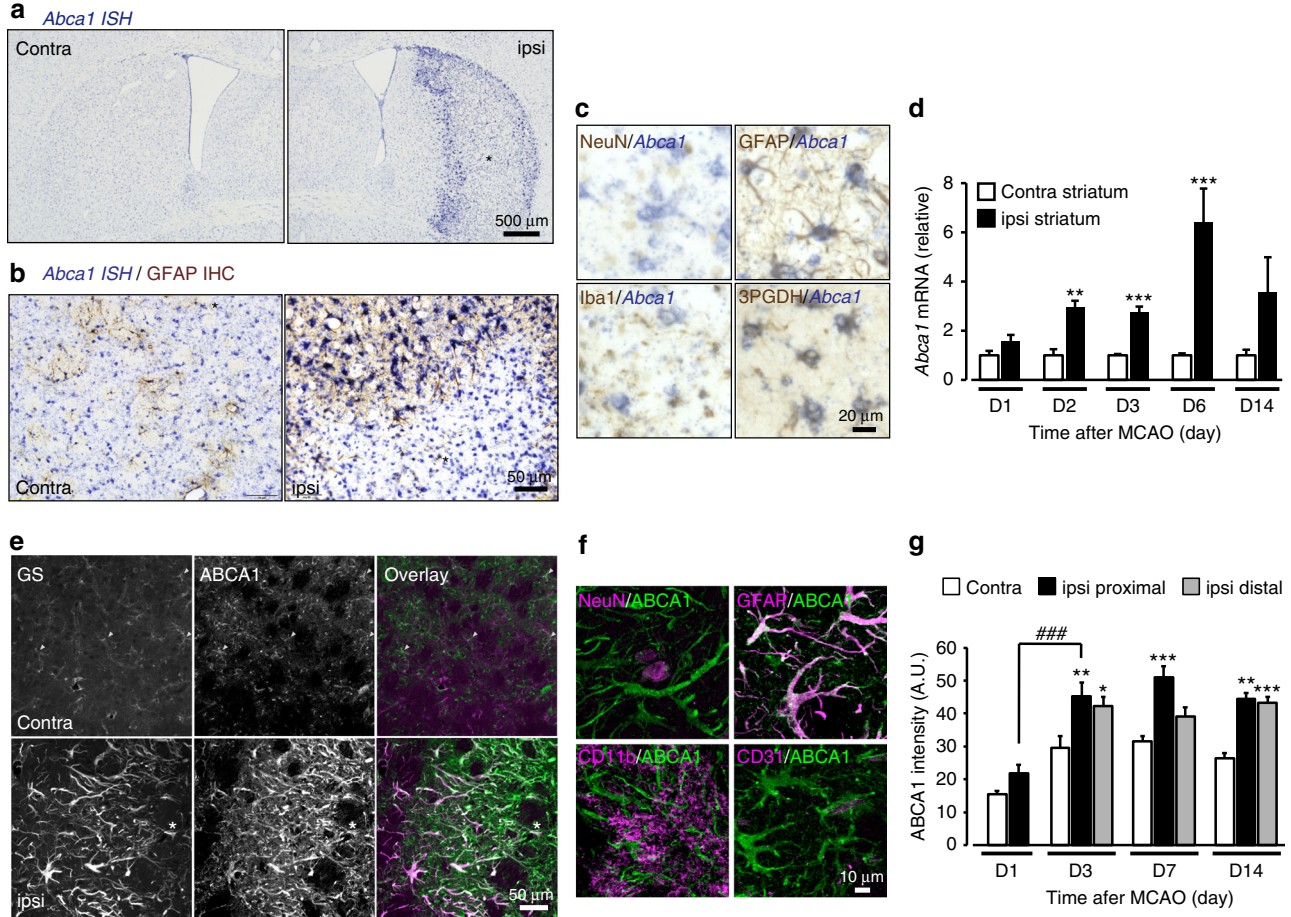

**Fig. 6** Transient ischemic injury induces ABCA1 upregulation exclusively in reactive astrocytes. **a** In situ hybridization (ISH) analysis of Abca1 mRNA at 7 days after MCAO. Abca1 ISH signals (*purple*) are strongly upregulated in the ischemic penumbra. **b** Representative images show that increased Abca1 ISH signals colocalize with GFAP immunoreactivity (DAB: *brown*). **c** Increased ABCA1 ISH signals colocalize with GFAP and 3PGDH, but not with NeuN and Iba1 in the ischemic penumbra at 7 days after MCAO. Low-magnification images from the contralateral side are shown in Supplementary Fig. 6b. **d** Real-time PCR analysis of Abca1 mRNA in total RNA extracted from the ipsilateral and contralateral striatum after MCAO. Values represent the relative ratio of Abca1 mRNA (normalized to GAPDH mRNA levels) to the corresponding contralateral striatum (D1: n = 4; D2: n = 3; D3: n = 8; D6: n = 9; D14: n = 6; *P < 0.05, **P < 0.01, ***P < 0.001 vs. contra (corresponding day)). **e** Immunostaining for ABCA1 and GS in the contralateral and ipsilateral striatum at 7 days after MCAO. Fifteen images per z-stack image (1.14 μm step). **f** Increased ABCA1 IHC signals colocalize with GFAP, but not with NeuN, CD11b, or CD31 in the ischemic penumbra at 7 days after MCAO. **g** Quantification of ABCA1 immunoreactivity mean intensity in GS+ astrocytes after MCAO (n = 9, 8, 12, 12, 21, 11, 12, 20, 12, 11, 21 fields, 3–4 mice, *P < 0.05, **P < 0.01, ***P < 0.001 vs. contra (corresponding day), ##P < 0.01 ipsi proximal D1, one-way ANOVA (P < 0.0001) with Tukey's multiple comparison test). *Asterisks* indicate the ischemic core. Values represent means ± SEM

the penumbra region directly adjacent to the ischemic core (within 200 μm from the ischemic core) where reactive astrocytes, microglia and FJ+ cells existed concurrently (7 days after MCAO). As shown in Fig. 3, Iba1+ microglia enwrapped many large FJ+ signals. In contrast, the processes of reactive astrocytes also enwrapped these signals, although the frequency was lower than for microglia (Fig. 3b, d). Most processes of astrocytes were adjacent to neuronal large debris and enwrapped numerous small FJ+ signals (Fig. 1d).

**Spatiotemporal characteristics of phagocytic astrocytes**. To characterize the spatiotemporal pattern of phagocytic astrocytes and microglia[11, 26, 27], we performed IHC analysis using Galectin-3 and lysosome markers (LAMP2 and CD68). Galectin-3+ cells were not detected in the contralateral striatum of the MCAO-injured brain. However, Galectin-3+ signals significantly increased in the ipsilateral striatum 1 day after MCAO, although the expression colocalized with Iba1+ cells and not GFAP+ cells (Fig. 4c). By 3 days after MCAO, Galectin-3+ cells were

significantly increased in both the core and penumbra regions of the ipsilateral striatum. The signals in the core region were colocalized with Iba1+ cells, whereas the signals in the penumbra region were mainly colocalized with GFAP+ reactive astrocytes (Fig. 4a). Galectin-3 expression in astrocytes peaked at 7 days after MCAO and lasted for at least 14 days after MCAO, after which expression gradually decreased (Fig. 4b). The majority of Galectin-3+ astrocytes was located in the rim of the penumbra region directly adjacent to the ischemic core and exhibited asymmetric and elongated processes (proximal astrocytes) (Fig. 4b). Galectin-3+ astrocytes were also present in the distal site of the core region (200–600 μm distance) (distal astrocytes), although the numbers were less than the proximal astrocytes. Similar to the Galectin-3 expression pattern, LAMP-2 immunoreactivity significantly increased in astrocytes (either GFAP+ or 3PGDH+: a pan-astrocyte marker[28–30]) within the ischemic penumbra (Fig. 5a). LAMP2 signals in astrocytes also peaked at 7 days and lasted for at least 14 days after MCAO (Fig. 5b). Conversely, increased of Galectin-3 and CD68 in Iba1+ microglia peaked at 3 days and mainly within the ischemic core, and did

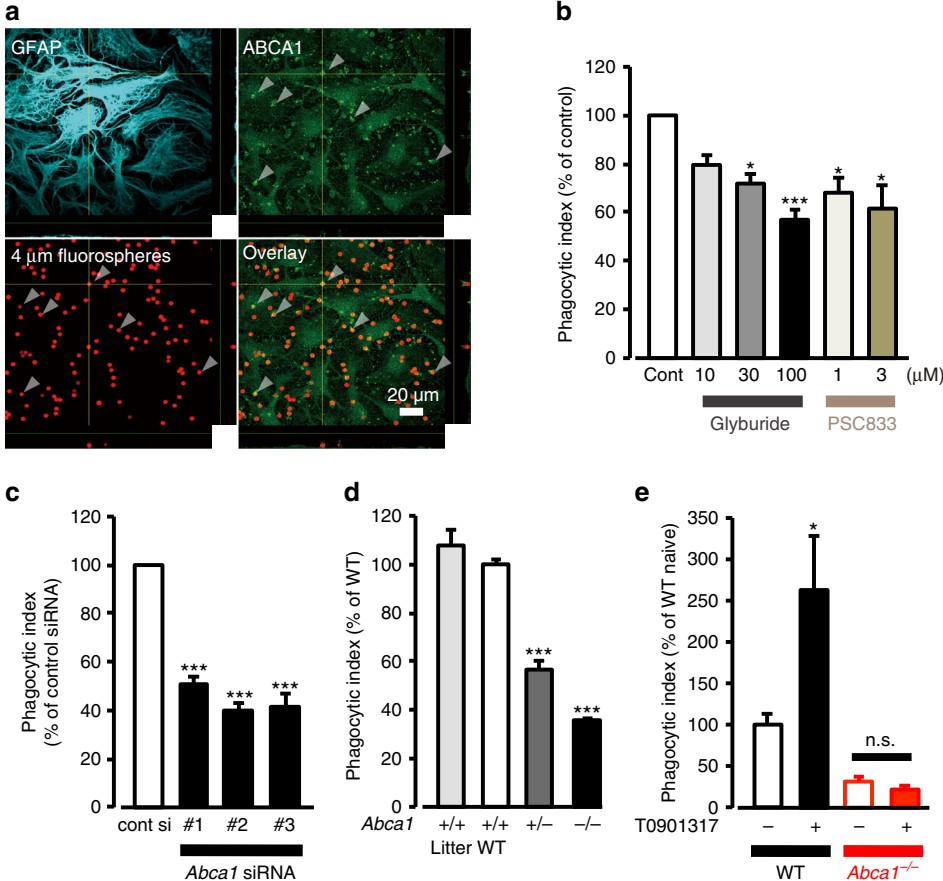

**Fig. 7** ABCA1 mediates astrocytic phagocytosis in vitro. **a** Confocal images show astrocytes (GFAP, *cyan*), ABCA1 (*green*), and 4-μm fluorospheres (*red*). Astrocytes were incubated with 4-μm fluorospheres (30 min after incubation). *Arrowheads* indicate ABCA1 accumulation and points of attachment between captured beads and astrocytes. Forty-eight images per z-stack images (0.38 μm step). **b** Phagocytic activities in the presence of ABCA1 inhibitors. Astrocytes were pretreated Glyburide or PSC833 for 15 min prior to the addition of 4-μm fluorospheres, and uptake was assessed by FACS after 1 h ($n = 15, 6, 6, 15, 6, 6$, $*P < 0.05$, $***P < 0.001$ vs. control, unpaired *t*-test). **c** Phagocytic activities of astrocytes transfected with control siRNA or Abca1 siRNAs ($n = 11, 14, 10, 11$, $***P < 0.001$ vs. control siRNA, unpaired *t*-test). **d** Phagocytic activities of astrocytes from wildtype (WT), Abca1$^{+/-}$, or Abca1$^{-/-}$ mice ($n = 3, 13, 7, 14$, $***P < 0.001$ vs. littermate control (Litter), one-way ANOVA ($P < 0.0001$) with Tukey's multiple comparison test). **e** Phagocytic activities of astrocytes from WT or Abca1$^{-/-}$ mice with or without T0901317 pretreatment (100 nM, 48 h) ($n = 11, 13, 7, 8$, $*P < 0.05$ vs. WT naïve, one-way ANOVA ($P < 0.0001$) with Tukey's multiple comparison test). Values represent means ± SEM for **b**–**e**

not last for 14 days after MCAO (Figs 4c, 5c, d). These data suggested spatiotemporal differences between phagocytic activity of astrocytes and microglia; i.e., the microglial phagocytosis has an early onset within the ischemic core, whereas the astrocytic phagocytosis has a late onset within the ischemic penumbra.

**Increased ABCA1 in reactive astrocytes**. To identify the molecules that drive astrocytic phagocytosis, we performed gene expression analysis for several engulfment pathway-related genes using tissues from the contralateral and ipsilateral striatum after MCAO. Real-time PCR results showed significantly increased *Abca1* mRNA expression in the ipsilateral striatum at 2 days compared with the contralateral side, which lasted for at least 14 days after MCAO (Fig. 6d). Although other genes were also upregulated (Supplementary Fig. 5), the extent of increased *Abca1* mRNA expression was most evident with high reproducibility among the engulfment pathway-related gene tested. Therefore, we focused on ABCA1 in this study. ABCA1 is known as the structural orthologue of *ced-7* and contributes to optimal engulfment of cell corpses in *Caenorhabditis elegans*[31–33]. In the periphery, ABCA1 also plays a pivotal role in the engulfment of apoptotic cells[34–40], although its role in engulfment in the central

nervous system remains poorly understood. To determine the expression pattern of *Abca1*, we performed in situ hybridization. *Abca1* expression was observed throughout the coronal section of the mouse brain and these signals were significantly upregulated (Abca1$^{high+}$) in the ischemic penumbra of the ipsilateral striatum after MCAO compared with the contralateral striatum (Fig. 6a). Combined with IHC staining, Abca1$^{high+}$ in situ hybridization signals colocalized with GFAP$^+$ or 3PGDH$^+$ astrocytes, but not with neuronal marker NeuN$^+$ or microglia Iba1$^+$ signals, in the ischemic penumbra (Fig. 6b, c; Supplementary Fig. 6). Consistent with in situ hybridization analysis, ABCA1 protein significantly increased in and colocalized with GS$^+$ or GFAP$^+$ reactive astrocytes, but not with NeuN, CD11b, or the vascular endothelial marker CD31 in the ischemic penumbra after MCAO (Fig. 6d–f). These results suggested that *Abca1* mRNA and ABCA1 protein significantly increased in reactive astrocytes after MCAO in the ischemic penumbra.

Previous studies have shown that ABCA1 plays a pivotal role in engulfment and cooperates with other molecules, such as MEGF10 (*ced-1*)[38], GULP1 (*ced-6*)[41], and Rac1 (*ced-10*)[42]. We analyzed the expression pattern of such molecules. In situ hybridization data showed that *Megf10* mRNA expression also increased in reactive astrocytes but not in Iba1$^+$ cells 7 days after

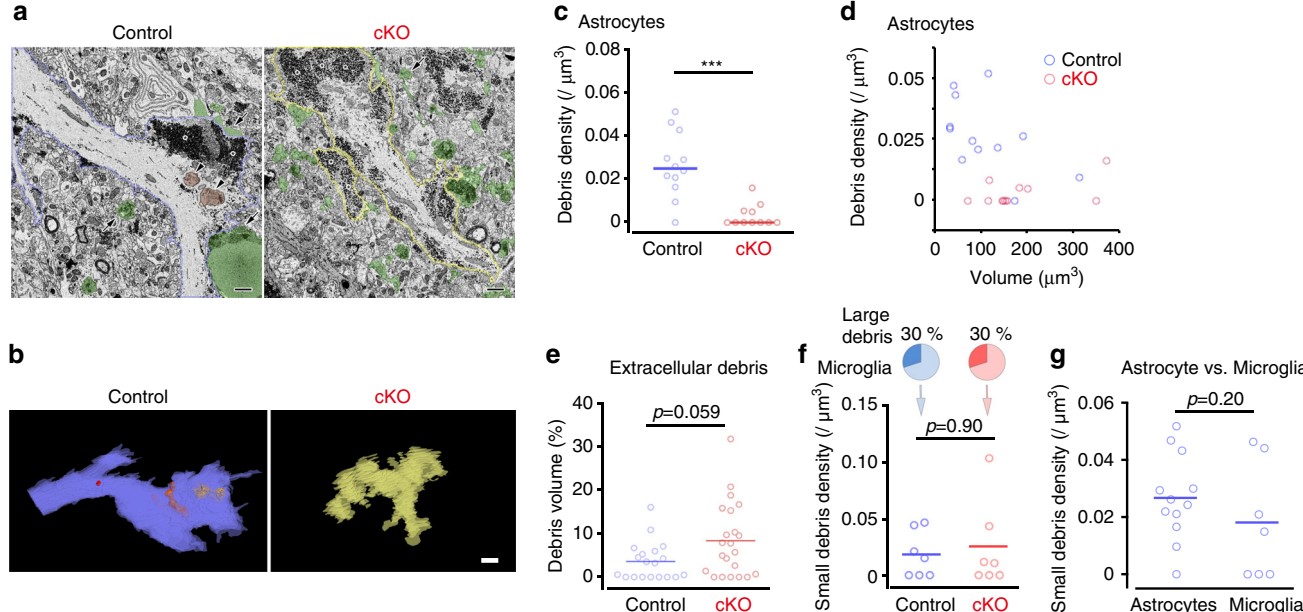

**Fig. 8** ABCA1 mediates astrocytic phagocytosis in vivo. **a, b** Serial electron microscopic images were acquired from the ischemic penumbra of control and ABCA1-cKO (cKO) mice, and three-dimensionally reconstructed (7 days after MCAO). Ischemic penumbra of control contains extracellular debris (**a**, *green*, *arrows*), and astrocyte processes (**a, b**, *left*, *blue*) with glycogen granules (**a**, *asterisks*) include cellular debris (**a, b**, *red*, *arrowheads*). By contrast, ischemic penumbra of cKO contains numerous extracellular debris (**a**, *green*, *arrows*), but the astrocyte processes (**a, b**, *yellow*) rarely contains debris. Bars: 1 μm **a** or 2 μm **b**. **c, d** Dot plots **c** and scatter plots **d** of debris densities (number/volume (μm³)) in astrocytes in control (*blue*) and cKO (*red*) mice (n = 12, 11 cells, three, four mice, respectively. ***P < 0.001 vs. control, Mann–Whitney U-test). **e** Dot plots show extracellular debris densities (%: volume/volume). The % volume occupied by extracellular debris tended to be higher in cKO, although the difference was not significant because of a large variance (n = 19, 22 regions, three, four mice, respectively. Mann–Whitney U-test). **f** Small debris densities in microglia are shown in a dot plot (n = 7 cells, three, four mice, respectively. Mann–Whitney U-test). These data excluded that microglia engulfed large cellular debris (maximum Feret diameter > 4 μm; control = 3, cKO = 3 cells). **g** Dot plots replotted from **c** and **f** show small debris densities in astrocytes and microglia in control mice (n = 12, 7 cells, three mice each. Mann–Whitney U-test)

MCAO (Supplementary Fig. 7), when many astrocytes had transformed into phagocytic cells, which is consistent with quantitative PCR data (Supplementary Fig. 5a). Consistently, IHC staining of MEGF10 increased in reactive astrocytes (Supplementary Fig. 8a). Although another cooperative molecule, GULP1, which was not changed in gene expression analysis significantly, was also upregulated in the ischemic penumbra and colocalized with GFAP⁺ signals in the ipsilateral striatum (Supplementary Fig. 8b). These results suggested that ABCA1-, MEGF10-, and GULP1-mediated pathways are involved in engulfment processes in reactive astrocytes after ischemic injury.

**ABCA1 pathway plays a pivotal role in astrocytic engulfment.** Recent work has shown that primary astrocytes cultured using a classical method[19, 43, 44] have many genetic and morphological similarities with reactive astrocytes induced by MCAO in vivo[21]. Thus, we investigated the involvement of ABCA1-mediated pathways in astrocytic phagocytosis using primary cultured astrocytes.

First, we determined whether cultured astrocytes ingested cell corpses and also characterized astrocytic phagocytosis in vitro. The cultured astrocytes were incubated with staurosporine-induced apoptotic neuronal debris labeled by the red fluorescent dye PKH26. The GFP-expressing astrocytes internalized or engulfed some of PKH26⁺ large signals, as well as many of the small fragments (Supplementary Fig. 9a, b). Astrocytes clearly formed a phagocytic cup structure, another hallmark of phagocytosis that consists of crown-like F-actin, when they engulfed the dead neuronal cell debris (Supplementary Fig. 9c). Furthermore, we observed engulfed neuronal cell debris that

colocalized with the LAMP1-GFP⁺ lysosomes (Supplementary Fig. 9d). Together these findings clearly showed that cultured astrocytes were phagocytic.

Astrocytic phagocytosis was assessed by FACS[11], and significantly decreased following exposure to reduced temperature (4 °C), the inhibitors of F-actin polymerization cytochalasin D, or the inhibitor of PI3-kinase LY294002 (Supplementary Fig. 9g). These characteristic phagocytosis features in astrocytes were similar to those observed in phagocytes. Furthermore, for easier quantitative analysis, fluorescent beads were used as a substrate for phagocytosis[11, 36]. Our results showed that the fluorescent beads could be successfully ingested as a pseudo substrate and the above-mentioned inhibitors inhibited uptake (Supplementary Fig. 9e, h). These inhibitors also decreased synaptosome uptake by astrocytes, indicating that cultured astrocytes have the ability to phagocytize apoptotic neurons, surrogate beads, and synaptic debris in our experimental model (Supplementary Fig. 9f, i).

To determine the involvement of ABCA1 in astrocytic phagocytosis, we initially used a pharmacological approach. Pretreatment of cultured astrocytes with Glyburide[34] or PSC833[45], which are functional inhibitors of ABCA1, significantly reduced uptake of fluorescent beads in a concentration-dependent manner (Fig. 7b). Additionally, when incubated with the fluorescent beads, ABCA1 polarized and localized with the beads as if ABCA1 surrounded the substrates (Fig. 7a). To directly determine whether endogenous ABCA1 is required for engulfment, we used three independent short interfering RNAs (siRNA) specific for Abca1 with distinct targets. All siRNAs specific for Abca1 significantly reduced astrocytic phagocytosis by approximately 50% compared with control siRNA, which was

associated with decreased *Abca1* mRNA and ABCA1 protein (Fig. 7c and Supplementary Figs. 10a, b, 16a, b). Moreover, we explored engulfment ability of astrocytes from *Abca1*[−/−] mice[35], with a targeted deletion of the ABCA1 gene in exon 17-22. *Abca1*[−/−] astrocytes had significantly less phagocytic ability toward beads and neuronal cell debris (Fig. 7d, Supplementary Fig. 10c, f), without showing any morphological abnormalities, actin-filament, or endocytotic activity (Supplementary Fig. 10c–e). Glyburide and PSC833 had no effect on Abca1-deficient astrocytes (Supplementary Fig. 10g), strongly supporting our hypothesis that ABCA1 and its functions are essential for astrocytic phagocytic activity.

We also demonstrated the involvement of other molecules in the ABCA1 pathway. MEGF10 and GULP1 were also upregulated in astrocytes in the ischemic penumbra after MCAO. Phagocytosis was decreased by almost 50% in the knockdown of either MEGF10 or GULP1 (Supplementary Fig. 10h, i). Because ABCA7, another ABC transporter family protein, has been shown to play a role in phagocytosis of apoptotic cells in macrophages[46], we evaluated the contribution of ABCA7 in this process. Results showed that ABCA7 knockdown did not decrease astrocytic phagocytosis (Supplementary Fig. 10h, i). These results suggested that endogenous ABCA1 and its pathway molecules MEGF10 and GULP1 are necessary for astrocyte engulfment in vitro.

**Upregulated ABCA1 increases astrocytic engulfment**. To determine whether upregulation of ABCA1 is sufficient to enhance astrocytic engulfment, we pharmacologically increased ABCA1. It is well known that the liver X receptor (LXR) and retinoid X receptor form a heterodimer that binds to the proximal promoter of the ABCA1 gene, resulting in increased gene transcription[47]. Treatment with the LXR agonist T0901317 increased both *Abca1* mRNA and ABCA1 protein in cultured astrocytes without strongly affecting mRNA expression of *Megf10*, *Gulp1*, *Abca7*, or *Mertk* (Supplementary Figs. 10h–l, 16b). Importantly, the increased ABCA1 following T0901317 treatment correlated with astrocytic phagocytosis (Fig. 7e, Supplementary Fig. 10m). Conversely, the T0901317-evoked increase in phagocytosis was not observed in ABCA1-deficient astrocytes (Fig. 7e, Supplementary Fig. 10m). Taken together, ABCA1 upregulation itself can enhance engulfment in cultured astrocytes.

**ABCA1 has a critical role in astrocytic phagocytosis in vivo**. Taken together, the preceding experiments provide evidence that engulfment mediated via the ABCA1-MEGF10-GULP1 pathway and ABCA1 upregulation itself enhanced phagocytosis in cultured astrocytes. To further explore this mechanism in vivo, we analyzed astrocyte-specific ABCA1 knockout mice (ABCA1[flox/flox].:: Cre-negative: control mice; ABCA1[flox/flox].:: GFAP-Cre: ABCA1 cKO mice) after MCAO[48]. There were no significant differences in the success rate of MCAO operation (both groups 50%, n = 12 and 14, respectively), with control and ABCA1 cKO mice showing reduced cerebral blood flow (CBF) during occlusion and a similar degree of MCAO-induced brain damage (Supplementary Fig. 11). In ABCA1 cKO mice, ABCA1 protein levels were decreased by 50% in the striatum and ABCA1 mRNA upregulation after MCAO was effectively abolished in reactive astrocytes (Supplementary Figs. 12a, b, 16c). Furthermore, cultured astrocytes from ABCA1 cKO mice showed an 80% reduction in ABCA1 protein levels and reduced phagocytic ability compared with controls (Supplementary Figs. 12c, d, 16d). We analyzed the phagocytic ability of astrocytes based on Galectin-3 and LAMP2 immunoreactivities and found that the intensity of both markers was significantly lower than in

littermate controls (Supplementary Fig. 13). To determine the exact phagocytic ability of astrocytes in vivo, we conducted 3D-EM analysis using SBF-SEM and showed that astrocytes sampled from ABCA1 cKO mice had significantly decreased numbers of phagosomes in their processes compared with littermate controls (Fig. 8a–d, Supplementary Fig. 14a). Consistent with a deficit in engulfment by astrocytes from ABCA1 cKO mice, extracellular debris tended to be higher in ABCA1 cKO tissues, although the difference was not significant because of a large variance between samples (Fig. 8e). Interestingly, astrocyte processes frequently touched or were very close to large debris (Figs. 2g, 8a), phagocytic inclusions in astrocytes were not so large (Fig. 8a). In contrast, microglia engulfed large debris (maximum Feret diameter >4 μm) and the phagocytosis of large and small debris by microglia was comparable between control and ABCA1 cKO mice (Fig. 8f, Supplementary Fig. 14b). The densities of engulfed small debris in astrocytes and microglia were comparable in control mice (Fig. 8g, mean debris density: astrocytes = 0.0267 μm$^{-3}$, microglia = 0.0180 μm$^{-3}$; 11/12 astrocytes had small debris; 3/10 microglia had large debris and 4/10 microglia had small debris), indicating that engulfment of debris by astrocytes was a frequent event after MCAO. Taken together, these data demonstrated that phagocytic astrocytes also actively participate in the remodeling of damaged tissues as well as microglia and that phagocytic signaling through ABCA1 is one main molecular mechanism by which reactive astrocytes engulf debris.

## Discussion
We demonstrated that adult astrocytes can function as phagocytes in a restricted spatiotemporal pattern after transient brain ischemia, and ABCA1 and its related pathway molecules play indispensable roles in this mechanism. Our results revealed the following points. (1) After transient brain ischemia, reactive astrocytes became phagocytic and engulfed a variety of debris, including fractions of degenerating neuronal cell debris, pre- and post-synapses, myelin and immune cell debris. (2) Phagocytic astrocytes were observed in spatially and temporally restricted patterns; i.e., they are mainly observed in the penumbra region during the later phase after injury. Conversely, phagocytic microglia were evident in the ischemic core region during the early phase after injury. (3) ABCA1 was selectively increased in reactive astrocytes in the penumbra region, and which was required for phagocytosis in vitro. The ABCA1 pathway molecules, MEGF10 and GULP1, were also involved in phagocytosis in vitro. (4) The increase in ABCA1 was sufficient for enhanced astrocytic phagocytosis in vitro. (5) Genetic disruption of ABCA1 in astrocytes resulted in deficient engulfment in vivo. (6) Engulfment of small debris by astrocytes was as frequent as in microglia after brain ischemia. Overall, we propose a new role for reactive astrocytes as phagocytes in pathophysiological condition. Astrocytes could work cooperatively with microglia and contribute to the engulfment of dead cells or debris. However, because there are differences in spatiotemporal phagocytosis patterns between astrocytes and microglia, astrocytes play a distinct role in clearance of damaged tissue from microglia. Results suggest that the phagocytic astrocytes likely contribute to remodeling and recovery of the brain microenvironment within the ischemic penumbra region (Supplementary Fig. 15).

EM analyses showed many phagosomes in astrocytes in the ipsilateral striatum after ischemic injury, but detected very few or no phagosomes in astrocytes in healthy or contralateral striatum, respectively (Figs. 1f, 2a, i), suggesting that astrocytic phagocytosis is not a frequent event in the healthy adult brain but rather occurs under pathophysiological conditions. These findings are in accordance with our IHC data that show Galectin-3[+] astrocytes

and LAMP2 signals in astrocytes were low in the healthy adult brain but significantly increased after the ischemia (Figs. 4, 5). However, previous studies reported that adult astrocytes in the healthy optic nerve head myelination transition zone actively phagocytose axonal debris[17, 18], and astrocytes in the cortex are involved in synapse phagocytosis[20]. This suggests that astrocytic phagocytosis in the healthy adult brain could occur in several brain regions, but its frequency seems to be partly brain-region-dependent and is low in the striatum. In contrast to array tomography analysis[20], our EM observations allowed the high-resolution analysis of a small area. This might explain why we failed to detect a low frequency astrocytic phagocytosis in the healthy adult striatum.

During development, astrocytes show intense phagocytic activity of synapses in the lateral geniculate nucleus[20]. Although these astrocytes might engulf synapses in the cortex of adult brains, but the efficacy is much lower than during developmental stages[20]. During brain development, network structures and functions dynamically change as well as after acute brain injury, and the brain becomes more plastic than normal adult brain under these situations. Accumulating evidence indicates both astrocytes and microglia play roles in remodeling during these dynamical plastic changes[7, 49–57]. Therefore, there might be functional and genetic similarities, under pathophysiological conditions, adult astrocytes might transform into a more phagocytic state similar to immature and/or developmental astrocytes. Together, these findings suggest that astrocytes can phagocytize adjacent materials, including synapses, although this occurs primarily during development or under pathophysiological conditions in the adult brain. Further investigation might indicate whether adult astrocytes also participate in network remodeling via engulfment under physiological situations such as learning and memory. Recent findings revealed that neuroinflammation and ischemia induced different types of reactive astrocytes termed A1 (harmful) and A2 (protective), respectively[21, 58]. In this literature, the focus was limited to the functions of A1 reactive astrocytes, which have deficient synapse engulfment in the developmental lateral geniculate nucleus. Although this literature have not examine the functions of A2 reactive astrocytes yet, the heterogeneous activations of astrocytes should be taken into account when discussing other brain diseases.

To date, two redundant pathways have been reported to have a role in phagocytosis[31, 42]. The first pathway includes MEGF10 (ced-1), GULP1 (ced-6), Rac1 (ced-10), and ABCA1 (ced-7) and is thought to participate in the recognition and engulfment of apoptotic cells. The second pathway includes CrkII (ced-2), Dock180 (ced-5), and Elmo1 (ced-12). Although several phagocytosis-related molecules have been described in astrocytes[15, 17, 19, 59], the present study showed that ABCA1, MEGF10 and GULP1 were also upregulated in reactive astrocytes after ischemic injury (Fig. 6, Supplementary Figs. 5–8). In fact, in cultures, suppression and overexpression of these molecules resulted in inhibition and facilitation of astrocytic phagocytosis, respectively (Fig. 7). Additionally, the knockdown of both Megf10 and Gulp1 significantly decreased astrocytic phagocytosis (Supplementary Fig. 9), suggesting that ABCA1 and its pathway molecules play critical roles in astrocytic phagocytosis.

ABCA1 is a member of the ABC transporter family and known to transport lipid species, such as cholesterol and phospholipid, across the membrane bilayer[60]. Although the exact function of ABCA1 in engulfment has not been clearly revealed yet, there are various roles proposed, e.g., as ABCA1 remodel membrane phospholipids[34, 35], inducing efflux lipid burden from engulfed debris[36, 37] and recruiting phagocytic receptors around cell corpses during engulfment[38–40]. It is not known how ABCA1

contributes to the astrocytic phagocytosis observed in the present study, but according to the notion observed above, it is possible that it is involved in the recognition or engulfment of debris. On the other hand, we did not clearly show the involvement of second pathway. According to Chung et al.[20], immature astrocytes engulf synapses via two phagocytic receptors, i.e., MEGF10 and MerTK; the former and the latter function as upstream phagocytic receptors for the first and the second pathways, respectively. Thus, astrocytic phagocytosis might partly share similar pathways during developmental stages and under pathological conditions.

Previous studies have suggested that astrocytes also express several other phagocytic receptors, including BAI1 and integrin αvβ3 or 5, which appear to function as upstream signals of the second pathway[59]. It remains poorly understood why phagocytic cells have multiple phagocytosis-related molecules, including phagocytic receptors and intracellular signaling molecules. Recent finding showed that astrocyte-like glial cells eliminate neuronal subcompartments via context-dependent use of distinct engulfment pathways during larval metamorphosis in Drosophila[61]. This suggests that Drosophila astrocytes use these molecules differently depending on their targets. If this is also the case for mammalian astrocytes, the ABCA1-dependent astrocytic phagocytosis observed in this study might be specific for special targets.

Unlike astrocytes, microglial engulfment was evident during the early phase within the ischemic core region (Figs. 4, 5), where many cells, including neurons, astrocytes, and other brain cells, undergo cell death. This rapid clearance of dying cells by professional phagocytes could prevent diffusion of detrimental contents owing to subsequent loss of permeability of the cell membranes. Results from previous studies suggested that the kinetics of phagocytosis are different between professional and non-professional phagocytes, i.e., professional phagocytes have higher rates and capacity for phagocytosis[62, 63]. Even under pathological conditions, astrocytes were not as mobile as microglia[26, 64] suggesting astrocytes might not be involved in the acute clearance of damaged tissue in the ischemic core region. In the present study, reactive astrocytes enwrapped many FJ+ small fractions (Fig. 1), but the enwrapment of FJ+ large neuronal cell bodies by reactive astrocytes was less frequent compared with microglia (Fig. 3). In addition, 3D-EM observations revealed no large phagocytic inclusions such as whole degenerating neurons in astrocyte cytoplasm although their processes were attached to large debris. Furthermore, astrocytic lysosomal signals (based on LAMP2 IHC) demonstrated the presence of smaller particles (Fig. 5). This suggests that there might be a limit to the size of debris that can be engulfed by astrocytes. 3D-EM also indicated that reactive astrocytes as well as microglia engulfed a large amount of small debris. The volume and number of astrocyte processes are much higher compared with microglia; therefore, the amount of debris engulfed by astrocytes would be substantial in the penumbra.

The onset of astrocytic phagocytosis began at 3 days and persisted for 2 weeks after MCAO within the ischemic penumbra region (Figs. 4, 5). Interestingly, LAMP2 upregulation in astrocytes was broader in areas distal from the ischemic core; however, CD68 upregulation in microglia was limited to the ischemic core and proximal region (Fig. 5). The spatiotemporal pattern of astrocytic phagocytosis suggests a relationship to neuronal remodeling in the ischemic penumbra region[7, 65]. There are substantial axonal, dendritic and synaptic losses and eliminations within the penumbra region within the first week after stroke, and this is followed by an increased number of synapses and axonal connections[6]. The present study demonstrated that synaptic debris became incorporated into reactive astrocytes in the

penumbra region, and that cultured astrocytes actively phagocytized synaptosomes (Supplementary Figs. 2, 9). These findings support our notion that reactive astrocytes also engage in synapse elimination in the penumbra region. However, we are not proposing that astrocytes are the only cell type that can eliminate synapses after ischemia. Microglia are known to eliminate synapses during developmental stages and pathological conditions[50, 51, 56, 57]. Further studies are needed to provide a better understanding of the physiological consequences of astrocytic phagocytosis, and to better elucidate the difference between phagocytosis in astrocytes and microglia.

In conclusion, we demonstrated that astrocytes become phagocytic after brain ischemia, and ABCA1 and its pathway molecules play a pivotal role in this process. The spatiotemporal pattern of astrocytic phagocytosis after stroke, i.e., slow onset and within the penumbra regions, suggests that astrocytes are involved in the elimination of debris and synaptic remodeling, and these mechanisms seem to be different from microglia. To date, studies have focused on the mechanisms involved in microglial phagocytic events, and astrocytes have received very limited attention. Results from this study provide novel information about reactive astrocytes and their role in phagocytic events following ischemic injury in the brain and can hopefully be applied to other brain diseases.

## Methods

**Animals**. Wistar rats, C57BL/6 and DBA1J mice were used for these studies. The $Abca1^{-/-}$ and Abca1$^{flox/flox}$::GFAP-Cre mice were available from a previous study; the generation and maintenance have been previously described in detail[35, 48]. Rats and mice were housed on a 12-h light (6 am)/dark (6 pm) cycle with ad libitum access to water and rodent chow. Mice were housed no more than five per cage. All experimental procedures were performed in accordance with the "Guiding Principles in the Care and Use of Animals in the Field of Physiologic Sciences" published by the Physiologic Society of Japan and with the previous approval of the Animal Care Committee of Yamanashi University (Chuo, Yamanashi, Japan).

**Transient focal ischemia**. Middle cerebral artery (MCA) occlusion reperfusion was carried out using male C57BL/6 mice (8–12 weeks old, 22–26 g) under 1.0% isoflurane anesthesia. The right common carotid artery was exposed through a midline incision, and MCA occlusion was achieved by inserting a nylon monofilament with a heat-blunted tip coated with silicon thread through the proximal external carotid artery into the internal carotid artery and up to the MCA (9 mm from the internal carotid/pterygopalatine artery bifurcation: 8 mm for Abca1$^{flox/flox}$ mice). MCA occlusion was maintained for 15 min (30 min for Abca1$^{flox/flox}$ mice), and experimental mice were sacrificed at the indicated day post-reperfusion. Body temperature was monitored by a rectal thermometer and maintained at 37 °C using a warm pad during surgery and until the animals were awake. The animals were scored for neurological deficits prior to reperfusion as follows: 0, no deficit; 1, flexion of the torso and contralateral forelimb for less than 3 s when lifted by the tail; 2, circling to the affected side when walking; 3, flexion of the torso and contralateral forelimb for more than 3 s when lifted by the tail; 4, contralateral forelimb weakness upon application of pressure to the side of the body; 5, circling to the affected side using only the forelimb on a spot; and 6, no spontaneous locomotor activity. We excluded animals from data collection and analysis when behavioral scores were <5.

Changes in cerebral surface blood flow were monitored by using a laser speckle blood flow imaging system (Omegazone, Omegawave, Tokyo, Japan), which obtains high-resolution, two-dimensional imaging and has a linear relationship with absolute CBF as described previously[66]. Recordings were performed through the skull under 1.2% (v/v) isoflurane anesthesia. For each recording, the skull surface was wiped with saline-soaked gauze. The CBF was measured in identically sized regions of interest (0.7 mm$^2$) located 2.5 mm lateral from the bregma. We calculated the CBF defined as follows: CBF = (ipsilateral CBF−background CBF)/(contralateral CBF−background CBF) × 100 (%). We excluded animals from data collection and analysis when CBF was under 15% or over 35%.

**Immunohistochemistry**. The mice were deeply anesthetized with pentobarbital and transcardially perfused with PBS containing 0.6% heparin (v/v, 1000 U mL$^{-1}$; Mochida Pharmaceutical, Tokyo, Japan) followed by 4% paraformaldehyde (PFA) in PBS. The brains were removed and post-fixed overnight, and then cryoprotected with 15 and 30% sucrose in PBS for 1 day each. The brains were frozen and coronal sections (16 μm) were cut using a cryostat (Leica). Sections were permeabilized with 0.2% saponin or 0.1–0.3% Triton X-100 (v/v) in PBS, blocked with 5% goat serum (v/v; Sigma) or Block Ace (DS Pharma, Tokyo, Japan) in PBS, and incubated

for 2 days overnight at 4 °C in primary antibodies. The sections were incubated in the following primary antibodies in blocking solution: monoclonal mouse anti-GFAP (1:2000; Millipore, MAB3402), polyclonal rabbit anti-GFAP (1:2000; Millipore, AB5804), monoclonal rat anti-GFAP (1:2000; Invitrogen, 13-0300), monoclonal mouse anti-NeuN (1:100; Millipore, MAB377), polyclonal rabbit anti-NeuN (1:500; Millipore, MABN140), polyclonal rabbit anti-3PGDH (1:1000; gift from Dr. M. Watanabe), polyclonal rabbit anti-ABCA1 (1:1000; Novus Biologicals, NB400-105), polyclonal rabbit anti-MEGF10 (1:200; Millipore, ABC10), polyclonal rabbit anti-GULP1 (1:100; Novus Biologicals, NBP1-84553), monoclonal rat anti-CD31 (1:100; BD Bioscience, 5502740), monoclonal rat anti-LAMP2 (1:250; Millipore, MABC40), monoclonal rat anti-Galectin-3 (1:500; Cederlane, CL8942AP), monoclonal mouse anti-MAP2 (1:1000; Millipore, MAB378), polyclonal rabbit anti-synapsin I (1:500; Millipore, AB1543), polyclonal rabbit anti-PSD95 (1:250; Cell Signaling, 2507), monoclonal mouse anti-GS (1:250; Millipore, MAB302), polyclonal rabbit anti-Iba1 (1:1000; Wako, 019-19741), monoclonal rat anti-CD11b (1:250; Exbio, 12-595), and monoclonal rat anti-CD68 (1:250; Serotec, MCA1957). The secondary antibodies were Alexa 405/488/546/647-conjugated goat anti-mouse/rabbit/rat or chicken anti-IgG (1:500; Invitrogen). Double labeling with in situ hybridization, horseradish peroxidase (HRP)-conjugated anti-rabbit or anti-mouse IgG (1: 200; Medical and Biological Laboratories) for secondary antibodies. The slices were mounted with Vectashield Mounting Medium (Vector Lab) and examined using a confocal laser scanning system (Olympus FV-1000 or Leica TCS SP8) or Keyence BIOREVO (BZ-9000).

**Fluoro-jade staining**. After IHC staining, brain slices were mounted onto slides and dried on a warmer at 50 °C. To avoid overquenching IHC signals, reaction time in 0.06% potassium permanganate solution was decreased from 15 min (manufacturer's instructions) to 5 min. The slides were rinsed for 1 min in distilled water and then transferred to the FJ staining solution. The reaction time and concentration of FJ solution were also decreased from 30 min and 0.001% to 10 min and 0.0001%, to avoid non-specific staining[67]. The slices were mounted with DPX mountant (Sigma). All reactions were gently shaken on a shaker.

**In situ hybridization**. In situ hybridization was performed using antisense riboprobes for *Abca1* (GenBank accession number NM_013454, full length: 1-6786 nt) and *Megf10* (GenBank accession number NM_001001979, full length: 1-3444 nt). DIG-labeled riboprobes were synthesized from each plasmid using in vitro transcription (DIG RNA Labeling Mix; Roche). Cryosections (16–20 μm) were fixed in 4% PFA, treated with Proteinase K (1 μg mL$^{-1}$; Merck) for 1 h following fixation in 4% PFA, acetylated, and then hybridized with DIG-labeled probes in hybridization buffer [50% formamide, 5× saline sodium citrate (SSC) (1× SSC: 0.15 M NaCl, 0.015 M sodium citrate in diethylpyrocarbonate-treated water), 200 μg mL$^{-1}$ yeast tRNA, 0.1 mg mL$^{-1}$ heparin, 1× Denhardt's solution, 0.2% Tween 20, 0.1% CHAPS, and 5 mM EDTA] overnight at 65 °C. Hybridized sections were washed three times with 1× SSC containing 50% formamide at 65 °C for 15 min (wash I) and 30 min (wash II), and with 0.1× SSC at 65 °C for 30 min (wash III). Then, sections were washed twice with maleic acid buffer (0.1 M maleic acid pH 7.5, 0.1% Tween 20, and 0.15 M NaCl) and incubated with alkaline phosphatase (AP)-conjugated anti-digoxigenin (anti-DIG) antibody (1:2000; Roche) in blocking buffer (0.5% skim milk in PBS) overnight at 4 °C. AP was visualized by staining with nitro-blue tetrazolium chloride (Roche) and 5-bromo-4-chloro-3-indolyl-phosphate (Roche) according to the manufacturer's instructions. For double staining, IHC was performed following in situ hybridization. Transmitted light images were taken with an Olympus BX53 microscope connected to a CCD camera (DP72; Olympus) or Keyence BIOREVO (BZ-9000).

**iEM analysis**. For single iEM analysis, frozen sections from MCAO and intact mice were incubated with a primary rabbit anti-GFAP antibody (1:200, Dako, Z0334) for 3 days at 4 °C followed by incubation with a nanogold-conjugated anti-rabbit secondary antibody (1:100; Invitrogen, N-24916) for 1 day at 4 °C. For double iEM, the sections were incubated with mouse anti-synaptophysin (1:500; Sigma, S5768) and rabbit anti-GFAP antibodies for 4 days at 4 °C followed by incubation with nanogold-conjugated anti-mouse (1:100; Invitrogen, A-24921) and HRP-conjugated anti-rabbit (1:100; Jackson ImmunoResearch, 111-035-144) antibodies for 1 day at 4 °C, and visualized with DAB solution (3,3′-diamino-benzidine tetrahydrochloride; Wako, 040-27001) with H$_2$O$_2$. After fixation with 2.5% glutaraldehyde for 10 min and silver enhancement with the HQ-Silver Kit (Nanoprobes Inc.) for 10 min at 25 °C in the dark, the sections were post-fixed with 1% OsO$_4$, dehydrated through an ethanol gradient (50–100%), followed by acetone and QY1 (butyl glycidyl ether), and embedded in Epon. Ultrathin sections (70 nm thick) were prepared and stained with 2% uranyl acetate and 80 mM lead citrate for 15 and 10 min, respectively. The sections were observed under a transmission EM (JEOL model 1230), and images were collected using the Digital Micrograph 3.3 (Gatan Inc.).

**SBF-SEM analysis**. Pieces of brain tissues fixed by the transcardial perfusion of buffered 4% PFA were immersed in 0.1 M PB containing 4% PFA and 0.5% glutaraldehyde (pH 7.4) at 4 °C overnight, and 200-μm-thick slices were cut with a

vibratome (VT-1000S, Leica). The brain slices were incubated with a primary antibody against GFAP (Santa Cruz) and secondary antibody conjugated with HRP. Immunoreaction products were visualized using diaminobenzidine substrate (Sigma), and additionally fixed with 4% PFA with 0.5% glutaraldehyde. Following observation under a light microscope and identification of the ischemic penumbra, which is typically located in the striatum near the corpus callosum and cortex, tissue pieces including the penumbra regions were collected for tissue preparation for SBF-SEM imaging. En bloc heavy metal staining was performed as reported previously with some modifications[68]. Briefly, tissues were washed with PBS, treated with 2% OsO$_4$ in 0.15% K$_4$[Fe(CN)$_6$] for 1 h on ice, filtered 0.1% thiocarbohydrazide for 20 min and 2% OsO$_4$ for 30 min at room temperature (RT). Tissues were then treated with lead aspartate solution at 70 °C for 30 min. Each of these treatments was followed by washing five times with double distilled water for 10–15 min. Tissues were dehydrated in a graded series of ethanol (60, 80, 90 and 95%, 5 min each), incubated with acetone dehydrated using a molecular sieve, a 1:1 mixture of resin and acetone, and 100% resin, Quetol 812 (Nisshin EM, Tokyo, Japan). The samples were placed in a mold with conductive resin containing 7% Ketjen black in Quetol 812[68], and cured at 70 °C overnight. Blocks from each group were trimmed and mounted on aluminum rivets with conductive glue (CW-2400, Circuitworks). The surfaces of the trimmed samples were treated with gold sputtering to increase conductivity, and imaged under various imaging conditions in Merlin or Sigma (Carl Zeiss) equipped with 3View (Gatan). Serial images obtained by ImageJ and FIJI software plugins, and segmentation and image analyses were performed in TrakEM2[69], Microscopy Image Browser[70] and Amira (FEI Visualization Science Group, USA).

**Quantitative real-time PCR.** Total RNA from cultured cells or tissues was extracted and isolated with Nucleospin RNA (Macherey-Nagel) in accordance with the manufacturer's instructions. Cultured cells were lysed directly with lysis buffer (RA1 in Nucleospin RNA) and dissected tissues from ipsi- and contralateral striatum from 2 mm-thick brain sections were diced and homogenized with lysis buffer. For quantitative analysis of all mRNAs expression, we used One Step PrimeScript RT-PCR Kit (Takara). All TaqMan probes and primers were obtained from Applied Biosystems by Life Technologies: rodent *Gapdh* (4308313), *mouse Abca1* (Mm00442646_m1), *Abca7* (Mm00497010_m1), *Bai1* (Mm01195143_m1), *Gulp1* (Mm00518428_m1), *Megf10* (Mm01257625_m1), *Mertk* (Mm00434920_m1), *Lrp1* (Mm00464608_m1), *Itgav* (Mm00434486_m1), *and Scarb1* (Mm00450234_m1).

**Western blotting.** Cells or tissues were extracted with radioimmunoprecipitation assay buffer (10 mM Tris/HCl, pH 8.0, 150 mM NaCl, 1 mM EDTA, 1% sodium deoxycholate, 1% Triton X-100, 0.1% SDS, and protease inhibitors cocktails) on ice and centrifuged at 1000×*g* for 10 min at 4 °C to remove cell debris. The supernatant was transferred to a new tube, and mixed with Laemmli sample buffer (4% SDS, 20% glycerol, 10% 2-mercaptoethanol, 0.004% bromophenol blue, and 0.125 M Tris HCl, pH 6.8). For detection of ABCA1, the preparations were not boiled to avoid aggregation. Then, the samples were subjected to a 5–10% polyacrylamide gel, and the proteins were electrophoretically transferred to polyvinylidene difluoride membranes (BioRad). The membranes were blocked with Block Ace (DS Pharma) and incubated with a polyclonal rabbit anti-ABCA1antibody (1:1000; Novus Biologicals) and a monoclonal mouse anti-b-actin antibody (1:10,000; Sigma) in Can Get Signal Solution 1 (TOYOBO) overnight at 4 °C. Antibodies were detected by incubating an HRP-conjugated secondary antibody (1:10,000; GE Healthcare) in Can Get Signal Solution 2 (TOYOBO) at RT for 1 h. The blots were detected using Super Signal West Femto Substrate (Thermo Scientific) and a LAS-4000 imaging system (Fujifilm). For densitometric quantification, the relative band density of ABCA1 to β-actin was quantified using ImageJ (US National Institutes of Health), and ABCA1 values were normalized to β-actin values for the loading control.

**Cell culture.** For primary astrocyte cultures, dissociated cerebral cortical cells from P0–1 rats or mice were plated in T75 flasks and grown in DMEM (Dulbecco's modified Eagle's medium, Gibco) containing 5% fetal bovine serum (*v*/*v*), 5% horse serum (*v*/*v*,) 100 U mL$^{-1}$ penicillin, and 100 mg mL$^{-1}$ streptomycin (Gibco) until they were confluent. To purify astrocytes from cortical cultures, the cells were subjected to 12 h of continuous shaking for 7–10 days after plating, and the detached cells were subsequently removed. The cultures were treated with trypsin and the disassociated cells were re-plated in 12-well plates or 8-well glass chambers at a density of 2–30,000 cells cm$^{-2}$. The 12-well plates and glass coverslips were coated with collagen and laminin (10 ng mL$^{-1}$). The medium was replaced every 3 days until the cells were confluent. At that time, the culture consisted mostly of astrocytes.

Dissociated neuronal cultures were prepared from E18 rat cerebral cortices and plated in 6-well plates at a density of 50,000 cells cm$^{-2}$. The 6-well plates were coated with polyethyleneimine (Sigma), and the cells were maintained up to 7 days in DMEM (Gibco) supplemented with 2% B-27 (*v*/*v*) (Gibco), 2 mM glutamine (Gibco), 100 U mL$^{-1}$ penicillin, and 100 mg mL$^{-1}$ streptomycin. Half of the culture medium was replaced every 3–4 days.

**In vitro phagocytosis assay.** Primary neurons were grown for 7 days, followed by transient exposure to staurosporine (100 nM) for 1 h and incubated for 5 h to induce apoptosis. We confirmed that neurons were induced to undergo apoptosis using fluorescein isothiocyanate -annexin V and, propidium iodide (PI) staining methods (early apoptosis (PI (−), annexin V (+)): 38.3 ± 10.8%; late apoptosis (PI (+), annexin V (+)): 31.1 ± 2.28%; live (PI (−), annexin V (−)): 17.8 ± 3.42%; *n* = 3, mean ± S.D.)). Synaptosomes were purified from adult mice brains by sucrose gradient multiple centrifugation. The mice were deeply anesthetized with pentobarbital and transcardially perfused with PBS containing 0.6% heparin (*v*/*v*, 1000 U mL$^{-1}$; Mochida Pharmaceutical). Dissected cortical tissues were diced and homogenized in HEPES/sucrose buffer (4 mM HEPES/0.53 M sucrose) and centrifuged at 800×*g* for 10 min at 4 °C The supernatant was then centrifuged at 9200×*g* for 15 min at 4 °C The pellet was then diluted with HEPES/sucrose buffer and this suspension was centrifuged at 10,200×*g* for 15 min at 4 °C. The pellet consisting of crude synaptosomes was resuspended in PBS. The prepared neuronal debris or synaptosomes were labeled by PKH26 red fluorescent dye (Sigma) in accordance with the manufacturer's instructions. Primary astrocytes were incubated with fluorescently labeled targets, i.e., PKH26-labeled apoptotic neurons, 2 μm carboxylate, and 4 μm sulfate-modified polystyrene fluorosphere (Molecular Probes). After the indicated time, the cells were extensively washed three times with cold PBS, incubated with 0.25% trypsin, and were resuspended in cold phosphate-buffered saline (PBS) containing 10% horse serum. The cells were analyzed using FACScalibur (BD Bioscience). Detached neuronal debris, fluorescent beads and synaptosomes were excluded for sampling using forward and side-scattered plots based on their homogeneity and small size compared with astrocytes. For each point, 5000–10,000 events were collected and the data was analyzed using Cell Quest. The phagocytic index was calculated by geometric mean of fluorescence intensity multiplied by the percentage of fluorescent-positive cells.

**Transfections.** We transfected plasmids and siRNAs into astrocytes at 3–5 days in culture with Lipofectamine 2000 (Invitrogen) according to the manufacturer's instructions. For gene silencing, we transfected 2 pmol of siRNA cm$^{-2}$. Prior to transfection, the astrocytes were fed fresh medium without antibiotics. Astrocytes were used at 72–96 h after transfection. All siRNAs were obtained from Applied Biosystems by Life Technologies: Silencer Select mouse *Abca1* siRNA #1 (s61785), #2 (s61786), #3 (s61787), *Megf10* (s88804), *Gulp1* (s206316), and *Abca7* (s77766) siRNA. Allstar negative control siRNA (1027281) was obtained from Qiagen. All siRNA sequences were non-disclosure.

**Immunocytochemistry.** Cultures were fixed with 4% PFA (wt vol$^{-1}$) in PBS for 30 min and then washed three times with ice-cold PBS. After blocking non-specific binding sites with PBS containing 10% Block Ace (DS Pharma) and 0.1% TritonX-100 for 1 h at RT, the cultures were incubated with primary antibodies in blocking buffer overnight at 4 °C. Primary antibodies were monoclonal mouse anti-GFAP (1:1000; Millipore, MAB3402), polyclonal rabbit anti-GFAP (1:1000; Millipore, AB5804), monoclonal mouse S100β (1:500; Santa Cruz, sc58841), polyclonal rabbit anti-ABCA1 (1:100; Novus Biologicals, NB400-105), and monoclonal mouse anti-synaptophysin1 (1:1000; Millipore, MAB5258). After washing the unbound antibody with three washes in PBS, the cultures were incubated with secondary antibodies for 1 h at RT. Secondary antibodies were Alexa 405/488/546-conjugated goat anti-mouse/rabbit/rat or chicken anti-IgG (1:500; Invitrogen). To visualize cell nuclei and F-actin, the cells were incubated for 30 min in a DAPI (1 μM; Dojindo) and phalloidin-Alexa 488 (6.6 μM; Invitrogen) solution, respectively. The cultures were washed three times with PBS and then fluorescent images were obtained using a confocal laser scanning system (FV-1000; Olympus).

**Image analysis.** Images were acquired using inverted confocal laser-scanning systems (Olympus FV-1000 or Leica TCS SP8) at 20×, 40×or 60× magnification, with a 0.75, 1.30 or 1.40 numerical aperture objective lens, respectively. Information about z-stack images was described in the figure legends. Images without descriptions are single plane images. The ischemic core was identified as the inside leading edge according to GFAP or 3PGDH immunoreactivity, whereas the penumbra region was defined as the region surrounding the core. For Galectin-3, LAMP2, CD68 and ABCA1 quantification analysis, striatal astrocytes were imaged based on 3PGDH or GS immunostaining in the contralateral striatum and the ipsilateral striatum at 1 and 3 days after MCAO. The remaining striatal astrocytes were imaged based on 3PGDH or GS immunostaining with GFAP immunostaining. Microglia were imaged based on Iba1 immunostaining. For each section, two or three fields were imaged in the defined space. Subsequent images were processed and quantified using FIJI (US National Institutes of Health; NIH). For these analyses, single plane images at 5 μm depth from the slice surface were used. To create a region of interest (ROI) for astrocytes, a threshold was set manually and particles were analyzed (size: 30–50 μm$^2$) based on 3PGDH, GS immunoreactivity. To create a ROI for microglia, a threshold was set manually and particles were analyzed (size: 10 μm$^2$) based on Iba1 immunoreactivity. To measure LAMP2, Galectin-3 and ABCA1 fluorescence intensity, only fluorescence within the astrocyte ROIs was analyzed. The numbers of Galectin-3$^+$ astrocytes with an S/N ratio ≥ 10.0 were quantified. Immunofluorescence intensity and cell counts were analyzed while

the investigator was blind to the experimental conditions. Figure 1c and Supplementary Fig. 3c were deconvoluted by HyVolution (Leica) to obtain confocal super-resolution images. For Fig. 3 analysis, a threshold was set manually and particles were analyzed (size: 10 μm²) to create ROI for large FJ⁺ signals. Processes of microglia or astrocytes were analyzed based on Iba1 or Gal-3, GFAP double-positive immunoreactivity, respectively. Three-dimensional surface rendering was performed by Surfaces (Imaris) based on GFAP immunoreactivity, and Syanpsin1 and PSD95 signals outside the astrocytic volume were subtracted by Mask properties (Imaris) in Supplementary Fig. 2. Colocalization analyses were performed with the Colocalization Threshold Image J plugin.

**Data analysis and statistics.** All statistical tests were run in Origin 9 (OriginLab Corp) or Prism7 (Graphpad). Data are presented as mean ± SEM. The number of imaging fields, animals used (*n*) is indicated in the figure legends, and there were always three or more separate animals for each experiment in all assays. Statistical analyses were performed with the unpaired *t*-test (two-sided), Mann–Whitney *U*-test, one-way ANOVA, and the post hoc Tukey–Kramer test.

**Data availability.** All relevant data are available from the corresponding author upon request.

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

## Acknowledgements

This work was supported by Grants-in-aid for Scientific Research (KAKENHI) on Innovative areas "Glial assembly" (25117003 to S.K.), on Challenging Exploratory Research (15K15524 to S.K.), on Research (B) (16H04669 to S.K.), Grant-in-Aid for JSPS Research Fellow (12J08505 to Y.M.), Grant-in-Aid for Young Scientists (B) (50772167 to Y.M.), Innovative Areas-Resource and technical support platforms for promoting research "Advanced Bioimaging Support" (JP16H06280), the Core Research for Evolutional Science and Technology Grant from the Japan Society for the Promotion of Sciences (to S.K. and J.N.), the Grant for the Cutting Edge Brain Sciences from Univ. Yamanashi, and Grant for Brain/MINDS from AMED (to S.S. and H.O.). We are grateful to Professor M. Hayden (British Columbia, Canada) for providing ABCA1^flox/flox::GFAP-Cre mice. We thank Associate Professor K. Matsui (Tohoku University, Japan) for helpful suggestions, Professor K. Ueda (Kyoto University, Japan) for critical comments, Dr K. Takanashi, Mr R. Komatsu, Mrs Y. Fukasawa and Mrs M. Tachibana (Univ. Yamanashi), Dr M. Horie, and Mr M.I. Hossain (Niigata Univ.) for technical assistance, and all members of the Koizumi Laboratory for critical discussion.

## Author contributions

Y.M.M. and S.K. conceived and designed the research. Y.M.M. performed most of the experiments and analyzed the data, and wrote the manuscript. Y.H. performed MCAO experiments. N.O. and Y.S. performed 3D EM experiments and analyzed the data. S.S. and H.O. performed iEM experiments and analyzed the data. K.S. and F.O. provided ABCA1-KO mice. H.T. contributed in situ hybridization experiments. Y.M., E.S., J.N. and S.K. analyzed/interpreted the data. S.K. supervised the project. All of the authors discussed and commented on the manuscript.

## Additional information

**Competing interests:** The authors declare no competing financial interests.

**Change history:** A correction to this article has been published and is linked from the HTML version of this paper.

