## [Peer Review File · Nature Communications]

Reviewers' comments:

Reviewer #1 (Remarks to the Author):

Reactive astrocytes function as a phagocyte after brain ischemia via ABCA1-mediated pathway
Yosuke Morizawa et al.

This manuscript for Yosuke Morizawa and colleagues aims to address the interesting topic of debris clearance and remodelling following ischemic injury in the cortex of adult mice. They start with a nice descriptive section using immunofluorescent microscopy, showing the different temporal location of phagocytic astrocytes and microglia - suggesting a different role for these two phagocytic cells in the central nervous system following injury. The hypothesis that astrocytes and microglia (both of which are competently able to phagocytose a range of different molecules under different conditions) may play different and non-redundant roles in clearance of damaged CNS cells following injury is intriguing - and provides a target for future research into possible pharmacotherapeutic intervention to help treat such traumatic injuries. Unfortunately, the remainder of the manuscript, though also novel in its approach, fails to address several key flaws in experimental design, which ultimately leave the conclusions of the paper to appear over-reaching and not founded on the data present here within. Specifically:

The main error of this manuscript is the reliance on a cell-culture proxy for in vivo phagocytosis - that is, after showing the engulfment of synapses in vivo (via staining with synaptophysin) the authors migrate to showing engulfment of neuronal debris in vitro, before then using only non-biological fluorescent molecules for the remainder of the study. The only reason given for such an approach is that it is 'easier to quantify' (line 224), which seems a poor proxy. This approach raises many questions:

Do astrocytes only engulf synapses in vivo? What about neuronal debris (their original in vitro test)? What about debris from other dying cells - eg. Oligo lineage and endothelial cells that would be damaged in such an ischemic injury?

Does the change in phagocytosis of the non-biological fluorescent molecule actually represent a biologically relevant alteration? Would this change in phagocytosis be seen in vitro if using biologically relevant neuronal debris or synapses? Could these pharmacological blockade experiments be conducted in vivo to see a change in phagocytic capacity?

Is the Mertk/Megf10/Abca1 pathway equally important for the engulfment of all targets by astrocytes? Are the differences for synapses, neuronal debris, myelin debris, etc.? Are the differences stated here simple a component of the engulfment of non-biological fluorescent molecules?

The authors state on several occasions that their astrocytes are able to engulf myelin debris (eg. Fig 1), however no experiment is designed to test this, nor is any data shown to validate these statements. They should be removed. Additionally, at line 288, they state that astrocytes are able to engulf immune cells - again with no data to substantiate these claims. Please remove.

Line 231 - do these astrocytes engulf more, or less, neuronal debris than activated astrocytes?

(line 69) - do these phagocytic astrocytes, once induced, ever return back to a 'non-phagocytic' phenotype? Or once transitioning into this state, do they remain?

Similarly, on line 235, do these pharmacological approaches cause a decrease in astrocyte reactivity transcripts? Do they alter other machinery within the cells, or only the phagocytic pathways?

Line 99 - STAT3 does not label all reactive astrocytes, only one of the (possibly many) activation states of these cells.

Figure 4, and line 268 - Abca1 expression has not been reported as upregulated in astrocytes following ischemic injury (the data provided here in Fig. 4 is of whole brain, with a mix of multiple cell types). How can the authors fit this known literature with their cell culture experiments in which they see an increase in this transporter? How appropriate is their cell culture model for interpretation of an astrocyte following ischemic injury?

In conclusion, the overall thesis of Morizawa et al., is highly interesting, and the pathophysiological role of such activated astrocytes is of great interest to not only the glial biology community, but also the wider neuroscience community as a whole. Unfortunately, the conclusions of this manuscript are not substantiated by the data provided, though if they were I would highly recommend this manuscript for consideration for publication. In its current state however, major revisions would be required to bring the manuscript up to the standards set by this journal.

Additional small comments:

Where there no phagocytic astrocytes during earlier stages of ischemic injury recovery? Similarly, where there no phagocytic microglia during later stages? These data are alluded to, but not specifically stated.

Line 53 - aside from resident microglia, what about the infiltration of other, peripheral professional phagocytes like macrophages or neutrophils, which are known to move into the CNS following such ischemic injuries (as well as other trauma).

Line 117 - I am not sure one can make comments about the process-bearing nature of astrocytes from EM images, without completing serial section reconstructions of cells. Can the authors provide any comment on the complexity of the processes - perhaps from IF or other low power light microscopy?

Line 135 - a possible discrepancy arises - as above it is stated that galectin-3 positive cells are GFAP positive astros? (as measured as FJ +ve cells?), while here galectin-3 signal is co-localised with Iba1 positive microglia.

Line 149 - the marker 3PGDH is not a common reactive astrocyte marker, and has not been introduced well to this section.

Line 154 - how long did the CD68 positive microglia persist following ischemic injury? There are reports in the literature of activated microglia being present several months following traumatic injury in the spinal cord.

Line 221 - what is the biological relevance of completing a phagocytosis assay at 40C?

Line 241 - what are these wild type astrocytes? Earlier in the manuscript the authors state that their cell culture model already produces cells that are like stroke-induced reactive astrocytes.

ABCA1 staining in Fig. 5a appears very high. Is there a baseline level of expression broadly across the cell surface of all astrocytes? - or is this simply non-specific staining of the antibody?

Line 327 - do the transcript signatures of the pathophysiological astrocytes resemble developmental/embryonic/immature astrocytes? The literature does not support this comment.

Line 336 - which Dock?

Line 336 - which Elmo?

Line 434 - two rectal thermometers?

Fig. 1 - state what the gold particles in immune-EM are labelling. Is it synaptophysin?

Fig. 4 - continuity, place boxes around all or no panels.

In general protein is not 'expressed', mRNA is expressed. This should be fixed throughout the manuscript.

Multiple continuity problems exist with gene and protein names. Ideally, genes should be italicised with a capitalised first letter (eg. *Gfap*, *Mertk*), while protein names should be capitalized (eg. GFAP, MERTK). This manuscript switches on several occasions - eg. *Mertk*, MERTK, MerTK.

There are several grammatical errors throughout the manuscript (eg. Line 50 '...thousands of debris...', line 196 'Although the another...') These should be carefully edited out of future versions of the manuscript.

Reviewer #2 (Remarks to the Author):

This manuscript reports several lines of evidence that in penumbra regions around forebrain stroke, certain reactive astrocytes become phagocytic and engulf debris from neurons, myelin and other cell types. The evidence presented includes identification of engulfed fluorescently labeled debris within immunoreactive astrocytes, also co-localized with the lysosomal marker LAMP2, and the phagocytosis associated molecule Galectin-3. In addition, immune-electron microscopy was used to provide ultrastructural evidence for engulfment of debris by astrocytes. Astrocyte engulfment of debris was found to peak around 7 days after stroke, but began around 3 days after stroke and continued for at least 14 days. Genetic regulation of astrocyte phagocytic activity was associated with ABCA1 pathway, the structural homolog of the *c. elegans* engulfment regulating gene, *ced-7*. Various loss- or gain-of-function experiments implicated the ABCA1 pathway in regulating astrocyte phagocytic activity.

The various experiments appear to have been well conducted and appropriately controlled. The data processing and statistical analyses seem appropriate and rigorous. The figures are of good quality. I found that the data convincingly support the interpretations presented in the text. The discussion is balanced and the text is well written.

Although evidence for phagocytic roles for astrocytes during development, in particular with regard to synapse pruning, at present there is little information available on potential roles of reactive astrocytes in phagocytic activities of debris or of synapses that might be dysfunctional. This study provides strong evidence that reactive astrocytes take part in such phagocytic activities in penumbra regions after stroke. The findings are likely to be of interest in the stroke field, and more broadly in other contexts of CNS damage or degeneration that might generate debris in need of phagocytosis and degradation. I have no major criticisms, but have one concern that requires attention.

Specific comments and concerns:

1. The first sentences of the Abstract and Discussion both state that "Adult astrocytes are typically quiescent...", but this is not correct, and it is not clear why the authors would want to make such a comment. Adult astrocytes are not quiescent but are very active in healthy tissue, where they exert numerous critical functions in response to many dynamic signaling events. The notion that astrocytes are "quiescent" in healthy tissue is antiquated and should not be perpetuated. This statement should be removed or edited in both the Abstract and Discussion. For example, "In addition to their functions in healthy tissue, astrocytes become reactive in response to various brain insults." There are many good reviews on dynamic astrocyte activity in healthy tissue that the authors may wish to look at.

Reviewer #3 (Remarks to the Author):

In the research article by Morizawa et al., the authors present complex in vitro studies and in vivo histological work demonstrating the role of astrocytes as phagocytic cells in cell cultures and in the brain after experimental cerebral ischemia. They also show that phagocytosis by astrocytes is partially mediated by the ABCA1 pathway. In general, the paper is interesting and the observations made could advance the understanding of the field. There are, however, some major issues which reduce the impact of the findings and should be considered by the authors.

The fact that astrocytes can phagocytose cell debris after ischemia is interesting. Phagocytosis by astrocytes has been reported previously, however the present findings are the first to show this in the ischemic brain. One major issue is that the functional role of phagocytic activity of astrocytes in neuronal loss, clearance of synapses, remodelling or overall outcome after cerebral ischemia has not been investigated in the paper. In other words, it remains unclear how much debris, synaptic structures, etc., is phagocytosed by astrocytes in the brain compared with the phagocytic activity of

microglia, macrophages or other leukocytes that are abundant in the brain 3-14 days after MCAo and it is also unclear whether blockade of astrocyte-mediated phagocytosis would change the injured brain milieu in any way in the present experimental model.

It would also be important to get an idea about the kinetics of astrocyte phagocytosis in vivo as in the paper only histological data suggest the association of cell debris with astrocytes, which are mostly outside the cells. Have the authors tried to follow these processes with live confocal or two-photon imaging? This could greatly increase the impact of the paper, as phagocytic activity is mostly devoted to microglia in the injured brain, which has been captured several times before using in vivo imaging.

The upregulation of ABCA1 in response to cerebral ischemia in astrocytes seems to be a convincing observation. However, while using multiple approaches to demonstrate that ABCA1 contributes to phagocytic activity of astrocytes in vitro, the authors did not make efforts to investigate this functionally in vivo. It could be done by either using KO mice or pharmacological approaches in order to demonstrate the relevance of these mechanisms in the injured brain.

Further points:

- Confocal microscopic analysis represents a key approach in the paper, but no information is provided regarding the number of Z planes recorded, the step size used and how subsequent analysis was performed. This should be described in detail. In many figures the authors show relatively low resolution images of very small particles to demonstrate phagocytic activity of astrocytes (e.g. Fig1a-f). Supplementary Fig. 3d and 3e are also good examples where colocalisation of Iba1-positive particles with LAMP2 is very difficult to interpret and no quantitative data is available about incidence of colocalisation between LAMP2 and Iba1. This should be provided.

- STAT3-mediated signal transduction is present in multiple cell types (neurons, microglia, macrophages, astrocytes), therefore STAT3 should not be used as a reactive astrocyte marker. It however could indicate glial - neuronal activation in response to injury. The authors should quantify STAT3 expression in different cell types in the brain after MCAo and discuss the results in the paper accordingly.

- It is not sufficient to show Fluoro-jade B (FJ) staining in astrocytes (Fig.1) whilst stating that they phagocytose neuronal debris after cerebral ischemia. FJ can also stain degenerating astrocytes and other cells (e.g. Damjanac et al., 2007; Anderson et al., 2003). The authors must perform co-detection of a neuronal marker with FJ and astrocytes at the same time to convincingly demonstrate phagocytosis of neurons and show quantitative data on how frequent astrocytic phagocytosis is compared to that of microglia. In addition, it has to be confirmed whether phagocytosis by astrocytes takes place independently of microglial phagocytosis i.e. FJ, Iba1 and GFAP staining should also be shown. This latter combination is also crucial since the vast majority of FJ staining is seen outside of astrocytes as shown in Fig.1b and it is similar in the experiment when fluorescently labelled apoptotic neurons were injected in the brain (Fig.1d). Showing galectin-3 staining in Fig.2. is not sufficient to represent the actual phagocytic activity of astrocytes and microglia. Higher resolution confocal images should also be shown for Figs.1-3.

- Supplementary Fig. 3: "We also found that Iba1+ signals localized with LAMP2+ lysosomal vesicles in the reactive astrocytes, indicating that phagocytic astrocytes engulf neuronal debris as well as immune cell debris." It is an interesting observation, but the majority of the Iba1 staining appears outside the astrocyte. The above statements have to be supported with higher resolution images and quantification, the present resolution and level of detail in the images is not sufficient to conclude.

- Fig.5a: "Arrowheads indicate ABCA1 accumulation and points of attachment between captured beads and astrocytes." There are no arrowheads in the panels.

- Details about how the authors assessed astrocyte phagocytosis by FACS should be shown in Supplementary Fig. 8. The authors should give examples to how they discriminated unbound cells or beads from astrocytes using forward and side-scatter and also explain this better in the methods. How did the authors assess the phagocytosis of synaptosomes in vitro? Did they prepare synaptosome fractions or just stained for synaptophysin after addition of PKH26 cells? If the latter, the amount of synaptophysin staining in astrocytes is difficult to interpret.

Responses to Reviewers comments

We thank all of the reviewers for their careful reading of our manuscript and their many helpful comments and suggestions for revision. We have worked hard to address all of the suggestions and have added many new experiments as well as text changes. A detailed response to each of the referee's comments is given below (referee's comments in black and our responses in blue).

Reviewers' comments:

Reviewer #1 (Remarks to the Author):

This manuscript for Yosuke Morizawa and colleagues aims to address the interesting topic of debris clearance and remodelling following ischemic injury in the cortex of adult mice. They start with a nice descriptive section using immunofluorescent microscopy, showing the different temporal location of phagocytic astrocytes and microglia - suggesting a different role for these two phagocytic cells in the central nervous system following injury. The hypothesis that astrocytes and microglia (both of which are competently able to phagocytose a range of different molecules under different conditions) may play different and non-redundant roles in clearance of damaged CNS cells following injury is intriguing - and provides a target for future research into possible pharmacotherapeutic intervention to help treat such traumatic injuries. Unfortunately, the remainder of the manuscript, though also novel in its approach, fails to address several key flaws in experimental design, which ultimately leave the conclusions of the paper to appear over-reaching and not founded on the data present here within.

Specifically: The main error of this manuscript is the reliance on a cell-culture proxy for in vivo phagocytosis - that is, after showing the engulfment of synapses in vivo (via staining with synaptophysin) the authors migrate to showing engulfment of neuronal debris in vitro, before then using only non-biological fluorescent molecules for the remainder of the study. The only reason given for such an approach is that it is 'easier to quantify' (line 224), which seems a poor proxy. This approach raises many questions:

Do astrocytes only engulf synapses in vivo? What about neuronal debris (their original in vitro test)?

Reply: We apologize that this part was confusing in the original manuscript. We showed that astrocytes engulfed endogenous degenerating neuronal debris labelled by Fluoro-jade B (FJ) as shown in the original version of Fig. 1a-c, e, f (except for original Fig. 1d). Furthermore, astrocytes internalized apoptotic neuronal debris injected exogenously (originally in vitro) in original Fig. 1d. However, the original Fig. 1d caused confusion and was unnecessary; therefore, we have deleted these data in the revised version of the manuscript.

Instead, we have added new data showing astrocytes enwrapped NeuN- and FJ-double positive degenerating neurons, and that NeuN⁺ signals colocalized with LAMP2 in astrocytes (revised version Fig. 1b, c). Reviewer #3 also pointed out the necessity of co-detection of a neuronal marker with FJ and astrocytes for the precise demonstration of phagocytosis of neurons. In addition, we have also provided new magnified images of GFAP, Gal-3 and FJ in revised Fig. 1d because higher resolution images were also requested by Reviewer #3. Furthermore, we have added new 3-dimensional electron microscopy (3D-EM) analysis that clearly shows astrocytes contain numerous phagosomes after MCAO (revised Fig. 2 and revised Supplemental Fig. 4.). Please also see below comments for Reviewer #3.

What about debris from other dying cells - eg. Oligo lineage and endothelial cells that would be damaged in such an ischemic injury?

As pointed out, astrocytes might engulf other dying cells including oligo lineage and endothelial cells. However, we did not find any evidence that oligodendrocytes and endothelial cells were engulfed by astrocytes when analysed by immunohistochemistry (IHC) combined with confocal microscopy. We used anti-oligodendrocyte (MAB 1580; Millipore) and anti-CD31 antibodies (AB312908 BioLegend) to stain oligodendrocytes and endothelial cells, respectively. However, both of these signals were not observed in GFAP⁺ astrocytes and even in Iba1⁺ microglia 7 days after MCAO (data not shown). Therefore, it is not clear whether astrocytes engulf these types of cells or that the detection of these cells engulfed by astrocytes might be difficult using IHC and confocal microscopy. Judging from the results that these cells were not detected even in microglia after MCAO, the latter would be more probable. Alternatively, antigens of oligodendrocytes or endothelial cells might disappear soon after cell death, damage or when engulfed by glial cells, and thus, IHC is not suitable for their detection. However, we must await further analysis to clarify this.

Does the change in phagocytosis of the non-biological fluorescent molecule actually represent a biologically relevant alteration? Would this change in phagocytosis be seen in vitro if using biologically relevant neuronal debris or synapses? Could these pharmacological blockade experiments be conducted in vivo to see a change in phagocytic capacity?

We should have explained that these fluorescent beads (carboxylate or sulfate-coated latex beads) are commonly used as simplified targets that mimic the negative-surface charge of apoptotic cells in phagocytosis research (Kiss et al., Curr Biol 2006; Erwig et al., PNAS 2006; Park et al., Nature 2007; Koizumi et al., Nature 2007). Indeed, the uptake of these beads was decreased by inhibitors of phagocytosis as for the uptake of apoptotic neuronal debris and synaptic debris (Supplemental Fig. 9g-i). In addition, we have newly added data that ABCA1-deficient astrocytes have less phagocytic ability for neuronal debris engulfment as well as fluorescent beads compared with wild-type astrocytes in the revised version of Supplementary Fig. 10g.

With regard to the use of pharmacological blockers in vivo, it is difficult to use actin polymerization or PI3K inhibitors because they have many non-specific and toxic effects in vivo. An ABCA1 inhibitor, Glyburide, which is an antidiabetic drug, could be used to inhibit the ABCA1-dependent phagocytosis of astrocytes after ischemic injury. However, it is easier to use astrocyte-specific ABCA1 knockout mice to manipulate ABCA1 in astrocytes in vivo, and thus, as shown in revised Fig. 8, we have prepared and used such conditional knockout mice. We have clearly shown the in vivo relevance of ABCA1 for the engulfment of debris by astrocytes after ischemic injury. Similar comments were also raised by Reviewer #3, and we have newly included several in vivo data in the revised manuscript (please see below, and new Fig. 8, Supplemental Figs. 11-14).

Is the Mertk/Megf10/Abca1 pathway equally important for the engulfment of all targets by astrocytes? Are the differences for synapses, neuronal debris, myelin debris, etc.? Are the differences stated here simple a component of the engulfment of non-biological fluorescent molecules?

We thank you for these interesting remarks. As mentioned in the Discussion, astrocyte-like glial cells eliminate neuronal subcompartments via the context-dependent use of distinct engulfment pathways during larval metamorphosis in Drosophila (reference 65). Although many scientists have

already investigated molecules involved in phagocytosis machineries, their specificities for targets, i.e., synapses, neuronal debris, and myelin debris, are poorly understood, and less is known about the mechanisms involved. Both MerTK and MEGF10 function as engulfment receptors by recognizing opsonin proteins bound to phosphatidylserine presented in target debris. MerTK recognizes Gas-6 and Protein S, and MEGF10 recognizes C1q and an unidentified molecule in mammals (Lew et al., eLife 2014; Iram et al., J Neurosci 2016; Wang et al., Nat Cell Biol 2010). Although the exact function of ABCA1 in engulfment is still unknown, it is required for the engulfment function of MEGF10 pathways (references 32-41 in the revised manuscript). As described above, we clearly showed that ABCA1 is important for the engulfment of debris both in vitro and in vivo (revised Figs. 7, 8). In this manuscript, we have focused on the fact that astrocytes become phagocytic after MCAO, and that ABCA1 is a key molecule in the process. Thus, we think that to clarify whether MerTK/MEGF10/ABCA1-pathways discriminate individual targets for their phagocytosis is slightly out of context for the current study and should be part of our next study.

The authors state on several occasions that their astrocytes are able to engulf myelin debris (eg. Fig 1), however no experiment is designed to test this, nor is any data shown to validate these statements. They should be removed.

Myelin structures can be observed under electron microscopy (EM), and were clearly detected in astrocytes after ischemic injury by immuno-EM (Fig. 1e, an arrow indicates myelin-like structure). We have newly added these data using 3D-EM to revised Fig 2h.

Additionally, at line 288, they state that astrocytes are able to engulf immune cells - again with no data to substantiate these claims. Please remove.

We have provided new clearer images and line profile status in revised Supplementary Fig. 3, showing the presence of Iba1⁺ fractions in LAMP2⁺ lysosomes in reactive astrocytes after ischemic injury.

Line 231 - do these astrocytes engulf more, or less, neuronal debris than activated astrocytes?

We think that cultured astrocytes we used are activated astrocytes. Previous transcriptome analysis of astrocytes cultured from neonatal brain by the classical

method (McCarthy and de Vellis, 1980) showed that they are very different from mature astrocytes in the healthy brain (Cahoy et al., 2008; Foo et al., 2011). As stated in the manuscript (lines 225-229), these cultured astrocytes have similar genetic features with reactive astrocytes after MCAO in vivo, and thus we think that these astrocytes show similar functions to activated astrocytes, i.e., these astrocytes would engulf neuronal debris as much as the activated astrocytes.

(line 69) - do these phagocytic astrocytes, once induced, ever return back to a 'non-phagocytic' phenotype? Or once transitioning into this state, do they remain?

Data in Figs. 4 and 5 (original Figs 2, 3) showed that phagocytic marker (Galectin-3 and LAMP2) immunoreactivities in astrocytes peaked at 7 days and then decreased significantly at 14 days after MCAO. This suggests that phagocytic astrocytes gradually return to a non-phagocytic phenotype.

Similarly, on line 235, do these pharmacological approaches cause a decrease in astrocyte reactivity transcripts? Do they alter other machinery within the cells, or only the phagocytic pathways?

We have not tested whether Glyburide and PSC833 affect astrocyte reactivity transcripts. However, these chemicals did not inhibit engulfment by ABCA1-deficient astrocytes indicating that they might inhibit ABCA1 functions leading to the inhibition of engulfment. We have added these results to Supplemental Fig. 10g.

Line 99 - STAT3 does not label all reactive astrocytes, only one of the (possibly many) activation states of these cells.

This was also raised by Reviewer #3. To avoid confusion, we have deleted these data from the revised version of the manuscript.

Figure 4, and line 268 - Abca1 expression has not been reported as upregulated in astrocytes following ischemic injury (the data provided here in Fig. 4 is of whole brain, with a mix of multiple cell types). How can the authors fit this known literature with their cell culture experiments in which they see an increase in this transporter? How appropriate is their cell culture model for interpretation of an astrocyte following ischemic injury?

We apologize for the confusing text in the original manuscript. We validated Abca1 mRNA and protein upregulation in astrocytes after MCAO using in situ hybridization and IHC staining in the revised Fig. 6 (original Fig. 4). Thus, we think that to increase ABCA1 in cultured astrocytes and to investigate its functional role in engulfment is appropriate. However, which molecules increase ABCA1 in astrocytes after ischemia is unknown. We use the LXR agonist T0901317 to increase ABCA1 in gain of function assay, i.e., whether the increase in ABCA1 enhances astrocytic engulfment. It is well known that LXR agonists induce Abca1 in astrocytes and other cells (Costet et al., JBC 2000; Venkateswaran et al., PNAS 2000; Chawla et al., Mol Cell 2001; Terwel et al., J Neurosci 2011). We did not use the LXR agonist T0901317 to mimic culture models of ischemic injury.

In conclusion, the overall thesis of Morizawa et al., is highly interesting, and the pathophysiological role of such activated astrocytes is of great interest to not only the glial biology community, but also the wider neuroscience community as a whole. Unfortunately, the conclusions of this manuscript are not substantiated by the data provided, though if they were I would highly recommend this manuscript for consideration for publication. In its current state however, major revisions would be required to bring the manuscript up to the standards set by this journal.

Additional small comments:

Where there no phagocytic astrocytes during earlier stages of ischemic injury recovery? Similarly, where there no phagocytic microglia during later stages? These data are alluded to, but not specifically stated.

These data were provided in Figs. 4 and 5 (original Figs 2, 3). We used the term 'phagocytic' based on the enhancement of phagocytic markers Galectin-3 and lysosomal protein LAMP2 (astrocytes) or CD68 (microglia) compared with the uninjured side (contra). We just stated there were more phagocytic microglia during earlier stages of ischemic injury and there were more phagocytic astrocytes during later stages based on these data.

Line 53 - aside from resident microglia, what about the infiltration of other, peripheral professional phagocytes like macrophages or neutrophils, which are

known to move into the CNS following such ischemic injuries (as well as other trauma).

This was stated and modified in Line 96 of the manuscript. As you pointed out, we confirmed that CD45^{high+} and CD11b^{high+} immune cell infiltration from the periphery occurred in our MCAO model (data not shown). However, we could not discriminate microglia from other peripheral immune cells in this study.

Line 117 - I am not sure one can make comments about the process-bearing nature of astrocytes from EM images, without completing serial section reconstructions of cells. Can the authors provide any comment on the complexity of the processes - perhaps from IF or other low power light microscopy?

Thank you for this comment. In the original manuscript, iEM data showed that GFAP-immunopositive astrocytes had phagocytic inclusions. As you stated, it is difficult to discriminate astrocyte fine processes from EM images without IHC staining or a serial sectioning approach. Thus, we have conducted and added 3D-EM data to revised Fig. 2 and Supplemental Fig. 4. In the stack of SBF-SEM (3D-EM) serial images, astrocytes were identified as cells, which were continuous from processes containing bundles of intermediate filaments and often surrounding basement membranes of blood vessels and/or synaptic connections.

Line 135 - a possible discrepancy arises - as above it is stated that galectin-3 positive cells are GFAP positive astros? (as measured as FJ +ve cells?), while here galectin-3 signal is co-localised with Iba1 positive microglia.

As shown in Fig. 4, during the earlier stages of ischemic injury (days 1 and 3 after MCAO), most Galectin-3⁺ cells were Iba1⁺ microglia in the ischemic core region. During the later stages (days 7 and 14 after MCAO), most Galectin-3⁺ cells were GFAP⁺ astrocytes especially in the penumbra region. Iba1⁺ microglia were also Galectin-3⁺ but only in the ischemic core region at day 7 after MCAO and the immunoreactivity was much lower than at day 3 after MCAO.

Line 149 - the marker 3PGDH is not a common reactive astrocyte marker, and has not been introduced well to this section.

We apologize for lack of explanation about 3PGDH. We used 3PGDH as a pan-astrocyte marker not a reactive astrocyte marker because the GFAP

antibody did not label astrocytes in the uninjured striatum. A description and reference have been provided in line 171 of the revised version of the manuscript (Furuya et al., PNAS 2000; Yamasaki et al., J Neurosci 2001; Ehmsen et al., J Neurosci 2013).

Line 154 - how long did the CD68 positive microglia persist following ischemic injury? There are reports in the literature of activated microglia being present several months following traumatic injury in the spinal cord.

Although Iba1⁺ microglia persisted over 2 months in the core following ischemic injury (data not shown), CD68 expression dropped to the level of uninjured striatum by 14 days after MCAO (revised Fig. 5).

Line 221 - what is the biological relevance of completing a phagocytosis assay at 40C?

The incubation of cells at a low temperature is a commonly used negative control for ingestion. In such conditions, phagocytes can bind targets, but the internalization of particles is largely prevented (Peterson et al., Infect Immun 1977; Mondal et al., Comp Biochem Physiol A Mol Integr Physiol 2001; Mao et al., JCB 2009). It is thought that low temperatures reduce membrane fluidity and influence many other pertinent cellular properties or activities.

Line 241 - what are these wild type astrocytes? Earlier in the manuscript the authors state that their cell culture model already produces cells that are like stroke-induced reactive astrocytes.

Thank you for indicating this error. "Wild type astrocytes" has been replaced with "non-treated astrocytes" in the revised version of the manuscript (Line 262).

ABCA1 staining in Fig. 5a appears very high. Is there a baseline level of expression broadly across the cell surface of all astrocytes? - or is this simply non-specific staining of the antibody?

As you stated, ABCA1 signals seem to be expressed broadly throughout astrocytes. In addition, when astrocytes bind to beads, they are localized at the bead binding sites. To analyse their spatial distribution more precisely, we performed additional experiments, and the expression of ABCA1-GFP was forced in cultured astrocytes. ABCA1-GFP signals were localized broadly in the surface membrane and at the bead binding sites (arrowheads) as shown below.

The distribution pattern of ABCA1-GFP was associated with ABCA1 as shown in Fig. 7a, and thus we think that ABCA1 signals are not simply non-specific staining of the antibody.

Line 327 - do the transcript signatures of the pathophysiological astrocytes resemble developmental/embryonic/immature astrocytes? The literature does not support this comment.

As you stated, the original manuscript was speculative. We have toned down the expression as follows.

Line 383

During brain development, network structures and functions dynamically change as well as after acute brain injury, and the brain becomes more plastic than normal adult brain under these situations. Accumulating evidence indicates both astrocytes and microglia play roles in remodelling during these dynamical plastic changes^{4, 56, 57, 58, 59, 60, 61, 62}. Therefore, there might be functional and genetic similarities, under pathophysiological conditions, adult astrocytes might transform into a more phagocytic state similar to immature and/or developmental astrocytes. Together, these findings suggest that astrocytes can phagocytize adjacent materials, including synapses, although this occurs primarily during development or under pathophysiological conditions in the adult brain.

Line 336 - which Dock?

Line 336 - which Elmo?

We apologize for the inadequate explanation. It is known that Crk II, Dock180 and Elmo1 mediate phagocytosis (Albert et al., Nat Cell Biol 2000; Gumienny et al., Cell 2001; Park et al., Nature 2007; Elliott et al., Nature 2010; Lu et al., Nat Cell Biol 2011). We have added these references and statements to the revised version of the manuscript (Line 407).

Line 434 - two rectal thermometers?

We thank you for noticing the duplication. We have deleted the redundant sentence in the revised manuscript.

Fig. 1 - state what the gold particles in immune-EM are labelling. Is it synaptophysin?

In Fig. 1, gold particles were used to label the GFAP antibody. Although we described this in Figure 1g and the legend in the original version of the manuscript, we have described this in the text body of the revised manuscript (Line 113).

Fig. 4 - continuity, place boxes around all or no panels.

We thank you for noticing this error. We have corrected this mistake in revised Fig. 6 (original Fig. 4).

In general protein is not 'expressed', mRNA is expressed. This should be fixed throughout the manuscript.

Multiple continuity problems exist with gene and protein names. Ideally, genes should be italicised with a capitalised first letter (eg. *Gfap*, *Mertk*), while protein names should be capitalized (eg. GFAP, MERTK). This manuscript switches on several occasions - eg. *Mertk*, MERTK, MerTK.

We thank you for the comment regarding the incorrect terminology and abbreviations. We have deleted "expression" when describing proteins, and have corrected all gene and protein names in the revised version of the manuscript.

There are several grammatical errors throughout the manuscript (eg. Line 50 '...thousands of debris...', line 196 'Although the another...') These should be carefully edited out of future versions of the manuscript.

We thank the reviewer for this comment. The revised manuscript has been edited and proofread by a native-English speaker.

Reviewer #2 (Remarks to the Author):

This manuscript reports several lines of evidence that in penumbra regions around forebrain stroke, certain reactive astrocytes become phagocytic and engulf debris from neurons, myelin and other cell types. The evidence presented includes identification of engulfed fluorescently labeled debris within immunoreactive astrocytes, also co-localized with the lysosomal marker LAMP2, and the phagocytosis associated molecule Galectin-3. In addition, immune-electron microscopy was used to provide ultrastructural evidence for engulfment of debris by astrocytes. Astrocyte engulfment of debris was found to peak around 7 days after stroke, but began around 3 days after stroke and continued for at least 14 days. Genetic regulation of astrocyte phagocytic activity was associated with ABCA1 pathway, the structural homolog of the *c. elegans* engulfment regulating gene, *ced-7*. Various loss- or gain-of-function experiments implicated the ABCA1 pathway in regulating astrocyte phagocytic activity.

The various experiments appear to have been well conducted and appropriately controlled. The data processing and statistical analyses seem appropriate and rigorous. The figures are of good quality. I found that the data convincingly support the interpretations presented in the text. The discussion is balanced and the text is well written.

Although evidence for phagocytic roles for astrocytes during development, in particular with regard to synapse pruning, at present there is little information available on potential roles of reactive astrocytes in phagocytic activities of debris or of synapses that might be dysfunctional. This study provides strong evidence that reactive astrocytes take part in such phagocytic activities in penumbra regions after stroke. The findings are likely to be of interest in the stroke field, and more broadly in other contexts of CNS damage or degeneration that might generate debris in need of phagocytosis and degradation. I have no major criticisms, but have one concern that requires attention.

Specific comments and concerns:

1. The first sentences of the Abstract and Discussion both state that "Adult astrocytes are typically quiescent...", but this is not correct, and it is not clear

why the authors would want to make such a comment. Adult astrocytes are not quiescent but are very active in healthy tissue, where they exert numerous critical functions in response to many dynamic signaling events. The notion that astrocytes are "quiescent" in healthy tissue is antiquated and should not be perpetuated. This statement should be removed or edited in both the Abstract and Discussion. For example, "In addition to their functions in healthy tissue, astrocytes become reactive in response to various brain insults." There are many good reviews on dynamic astrocyte activity in healthy tissue that the authors may wish to look at.

We thank you for these kind comments and pointing out the inappropriate description of healthy astrocytes in the adult brain. In the revised manuscript, we have now replaced "Adult astrocytes are typically quiescent..." with Astrocytes become reactive following various brain insults; however, the functions of reactive astrocytes are poorly understood." in the Abstract and deleted the sentence in the Discussion.

Reviewer #3 (Remarks to the Author):

In the research article by Morizawa et al., the authors present complex in vitro studies and in vivo histological work demonstrating the role of astrocytes as phagocytic cells in cell cultures and in the brain after experimental cerebral ischemia. They also show that phagocytosis by astrocytes is partially mediated by the ABCA1 pathway. In general, the paper is interesting and the observations made could advance the understanding of the field. There are, however, some major issues which reduce the impact of the findings and should be considered by the authors.

The fact that astrocytes can phagocytose cell debris after ischemia is interesting. Phagocytosis by astrocytes has been reported previously, however the present findings are the first to show this in the ischemic brain. One major issue is that the functional role of phagocytic activity of astrocytes in neuronal loss, clearance of synapses, remodelling or overall outcome after cerebral ischemia has not been investigated in the paper. In other words, it remains unclear how much debris, synaptic structures, etc., is phagocytosed by astrocytes in the brain compared with the phagocytic activity of microglia, macrophages or other leukocytes that are abundant in the brain 3-14 days after MCAo and it is also unclear whether blockade of astrocyte-mediated phagocytosis would change the injured brain milieu in any way in the present experimental model.

We agree with your comments and understanding the functional aspects of phagocytosis by astrocytes is of great interest to us. We have performed additional in vivo experiments to address these issues in the revised version of the manuscript.

1. Reactive astrocytes enwrapped many FJ⁺ small fractions (Fig. 1d), but the enwrapment of FJ⁺ large neuronal cell bodies by reactive astrocytes was less frequent compared with microglia (Fig. 3). However most processes were very close to the large debris.

2. 3D-EM analysis clearly showed that reactive astrocytes frequently contained small debris in their cytoplasm (20/22 observed processes had engulfed debris, Figs. 2, 8, Supplemental Fig. 4).

3. 3D-EM analysis showed that 30% of observed microglia had large debris (maximum Feret diameter > 4 μ m, Supplemental Fig. 14b). However, the processes of observed astrocytes contained no large debris, although several

processes were attached to large debris (Fig. 2g, 8a). Astrocytes appear to engulf mainly small cellular debris or possibly may break large debris into smaller ones and then engulf them.

4. Engulfed small debris densities in microglia and astrocytes were comparable, indicating that astrocytic phagocytosis was as frequent as microglia phagocytosis (Fig. 8g).

5. To confirm the importance of ABCA1 in astrocytic phagocytosis in vivo, we generated astrocyte-specific ABCA1 knock out (ABCA1 cKO) mice, and demonstrated that ABCA1-deficient astrocytes in vivo had less phagocytic ability after MCAO (Fig. 8a-d, Supplemental Figs. 13, 14, please also see a below comment in detail).

6. Consistent with a deficit in engulfment by astrocytes from ABCA1 cKO mice, extracellular debris tended to be higher in ABCA1 cKO tissue, although the difference was not significant because of the large variance (Fig. 8e).

Because astrocyte processes are very complex and too thin to quantify precisely, and a comparison of the total amount of engulfed debris using light microscopy might overestimate or underestimate values, we conducted 3D-EM analysis to compare and show how much debris was engulfed by astrocytes and microglia (new Fig 8).

With regard to the outcome of astrocytic phagocytosis, we are now investigating the functional roles and consequences by histological and behavioural long term approaches using astrocyte-specific ABCA1 cKO mice, whose astrocytes are less phagocytic after MCAO. However, we have not reached any conclusion yet, and it will take a long time to finalize these results. Thus, we think that these issues are likely to be solved in our next study.

It would also be important to get an idea about the kinetics of astrocyte phagocytosis in vivo as in the paper only histological data suggest the association of cell debris with astrocytes, which are mostly outside the cells. Have the authors tried to follow these processes with live confocal or two-photon imaging? This could greatly increase the impact of the paper, as phagocytic activity is mostly devoted to microglia in the injured brain, which has been captured several times before using in vivo imaging.

We agree that imaging of astrocytic phagocytosis would make our study more impressive and convincing. We have tried to perform these experiments in acute slices with two-photon microscopy, but so far we have not succeeded in imaging

of astrocytic phagocytosis because of technical difficulties. We are now actively working on this with new genetically encoded fluorescent probes to visualize the fine processes of astrocytes and degenerating neurons. It will take a long time to achieve these results, and we feel it would be the basis of an independent paper. Thus, we would like to leave for our next study.

The upregulation of ABCA1 in response to cerebral ischemia in astrocytes seems to be a convincing observation. However, while using multiple approaches to demonstrate that ABCA1 contributes to phagocytic activity of astrocytes in vitro, the authors did not make efforts to investigate this functionally in vivo. It could be done by either using KO mice or pharmacological approaches in order to demonstrate the relevance of these mechanisms in the injured brain. *We agree that astrocytic phagocytosis should be validated in vivo. Therefore, we performed additional in vivo experiments by 3D-EM analysis and immunohistochemical (IHC) staining. As shown above, we compared the phagocytic activity of astrocytes in $Abca1^{flox/flox}::Cre$ -negative mice (control) and $Abca1^{flox/flox}::GFAP-Cre$ mice (astrocyte-specific ABCA1 conditional knockout: cKO) (Supplemental Figs. 11, 12). Using 3D-EM analysis, we found that astrocytic processes in the ipsilateral striatum of WT mice engulfed a large amount of debris after MCAO (Fig. 2, Supplemental Fig. 4), whereas those in ABCA1 cKO mice engulfed less debris (Fig. 8a-b). In addition, using IHC analysis, we also found that reactive astrocytes in ABCA1 cKO mice had significantly less LAMP2 and Galectin-3 signals compared to control mice after MCAO (Supplemental Fig. 13). We have added these results to new Fig. 2, Supplemental Fig. 4, Fig. 8a-b, Supplemental Fig. 13 and the corresponding text in the revised version of the manuscript.*

Further points:

- Confocal microscopic analysis represents a key approach in the paper, but no information is provided regarding the number of Z planes recorded, the step size used and how subsequent analysis was performed. This should be described in detail. In many figures the authors show relatively low resolution images of very small particles to demonstrate phagocytic activity of astrocytes (e.g. Fig1a-f).

Supplementary Fig. 3d and 3e are also good examples where colocalisation of Iba1-positive particles with LAMP2 is very difficult to interpret and no quantitative data is available about incidence of colocalisation between LAMP2 and Iba1. This should be provided.

Thank for pointing out these inadequacies. We have added information about the Z planes and step size used in all revised figure legends. We have also added an explanation of further analyses performed in the "Image analysis" section of the Methods in the revised manuscript. We apologize for the low-resolution images. Our figure resolutions were originally not low, and therefore we have changed the method of uploading and provided greater magnified images in the revised manuscript.

With regard to Supplemental Fig. 3, we recaptured and reanalysed the colocalisation of Iba1⁺ particles with LAMP2 in astrocytes and have added quantitative data showing colocalisation between Iba1 and LAMP2 signals in GFAP⁺ astrocytes (Supplemental Fig. 3) in the revised manuscript.

- STAT3-mediated signal transduction is present in multiple cell types (neurons, microglia, macrophages, astrocytes), therefore STAT3 should not be used as a reactive astrocyte marker. It however could indicate glial - neuronal activation in response to injury. The authors should quantify STAT3 expression in different cell types in the brain after MCAo and discuss the results in the paper accordingly.

This was also raised by Reviewer #1, and we have deleted the data showing increased STAT3 in astrocytes because this was confusing and unnecessary for our point in the current study.

- It is not sufficient to show Fluoro-jade B (FJ) staining in astrocytes (Fig.1) whilst stating that they phagocytose neuronal debris after cerebral ischemia. FJ can also stain degenerating astrocytes and other cells (e.g. Damjanac et al., 2007; Anderson et al., 2003). The authors must perform co-detection of a neuronal marker with FJ and astrocytes at the same time to convincingly demonstrate phagocytosis of neurons and show quantitative data on how frequent astrocytic phagocytosis is compared to that of microglia. In addition, it has to be confirmed whether phagocytosis by astrocytes takes place independently of microglial phagocytosis i.e. FJ, Iba1 and GFAP staining should also be shown. This latter combination is also crucial since the vast majority of FJ staining is seen outside

of astrocytes as shown in Fig.1b and it is similar in the experiment when fluorescently labelled apoptotic neurons were injected in the brain (Fig.1d). Showing galectin-3 staining in Fig.2. is not sufficient to represent the actual phagocytic activity of astrocytes and microglia. Higher resolution confocal images should also be shown for Figs.1-3.

Thank you for these important comments. We have conducted IHC with FJ staining and shown that astrocyte processes enwrap NeuN- and FJ-double positive signals (Fig. 1b). Also, NeuN⁺ small fractions in astrocytes colocalized with LAMP2 signals in astrocytes (Fig. 1c).

We knew the previous studies showing that FJ could stain reactive astrocytes and microglia, and therefore, we carefully conducted FJ staining so as not to stain reactive astrocytes by using a lower FJ concentration and shorter reaction time. We have added these points to the Methods section in the revised version of the manuscript as follows;

Line 574

The reaction time and concentration of FJ solution were also decreased from 30 min and 0.001% to 10 min and 0.0001%, to avoid non-specific staining.

Using the procedure above, we clarified that GFAP⁺ reactive astrocytes did not have FJ-positive patterns, as seen in degenerating neurons and debris (Fig. 1, 3). Furthermore, our analysis showed that 100% of FJ⁺ large signals were colocalized with NeuN⁺ neuronal signals (Supplemental Fig. 1d). We have added these results to the revised version of the manuscript (Line 91).

In revised Fig. 3, we have shown the frequency of astrocyte processes enwrapping FJ⁺ large debris compared with microglia by IHC analysis. However, these analyses along with confocal microscopy and cellular markers is limited for detailed studies. Thus, we performed 3D-EM studies and compared the phagocytic abilities of astrocytes and microglia. Using this method, we found that astrocytic engulfment of debris was as frequent as microglia engulfment (small debris; maximum Feret diameter < 4 μ m). We have added these results to revised Fig. 8g.

IHC staining of Iba1 and GFAP with FJ staining were shown in Fig. 3a. Based on these data, we found that astrocytic phagocytosis occurred, at least in part, independently of microglial phagocytosis: (1) GFAP⁺ astrocytic processes enwrapped FJ⁺ large signals independently of microglia; (2) GFAP⁺ processes and Iba1⁺ processes enwrapped FJ⁺ large signals exclusively, and (3) GFAP⁺

processes enwrapped Iba1⁺ processes including FJ⁺ large signals. However, we need to clarify these points in more detail in the future.

- Supplementary Fig. 3: "We also found that Iba1⁺ signals localized with LAMP2⁺ lysosomal vesicles in the reactive astrocytes, indicating that phagocytic astrocytes engulf neuronal debris as well as immune cell debris." It is an interesting observation, but the majority of the Iba1 staining appears outside the astrocyte. The above statements have to be supported with higher resolution images and quantification, the present resolution and level of detail in the images is not sufficient to conclude.

We agree with this comment. As stated above, we conducted IHC analysis and have now provided high quality images showing many Iba1⁺ fractions were within GFAP⁺ astrocytes, some of which were colocalized with LAMP2. We have added these data and quantitative analysis to the revised version of the manuscript (Supplemental Fig. 3).

- Fig.5a: "Arrowheads indicate ABCA1 accumulation and points of attachment between captured beads and astrocytes." There are no arrowheads in the panels.

Thank you for pointing out this error. We have added arrowheads to revised Figure 7a.

- Details about how the authors assessed astrocyte phagocytosis by FACS should be shown in Supplementary Fig. 8. The authors should give examples to how they discriminated unbound cells or beads from astrocytes using forward and side-scatter and also explain this better in the methods. How did the authors assess the phagocytosis of synaptosomes in vitro? Did they prepare synaptosome fractions or just stained for synaptophysin after addition of PKH26 cells? If the latter, the amount of synaptophysin staining in astrocytes is difficult to interpret.

We apologize for the lack of explanation and reference regarding the *in vitro* phagocytosis assay and FACS analysis. As shown below, we easily distinguished apoptotic neuronal debris and fluorescent beads from cultured astrocytes based on their homogeneity and small size using forward (FSC; horizontal) and side-scatter plots (SSC; vertical). Signals inside the gates are detached targets; therefore, we analysed signals outside the exclusion gate.

With regard to synaptosome engulfment, we prepared a synaptosomal fraction from mouse brain by multiple centrifugations and labelled cells with PKH26. Similar to neuronal debris, we incubated these PKH26⁺ synaptosomal fractions with cultured astrocytes and quantified their phagocytic ability by FACS. To confirm that PKH26⁺ signals were synaptic debris, we performed synaptophysin 1-staining (Supplemental Fig. 9f). We have added explanations to the revised version of the Methods (Line 707).

REVIEWERS' COMMENTS:

Reviewer #1 (Remarks to the Author):

specific responses to author rebuttals are included in the attached word document (red text)

Additional comments on new material:

Line 398 - how have the authors determined that the astrocytes they are studying in this study are 'A1' or 'A2'? There does not appear to be any sequencing data to validate this claim.

Line ~462 - Comments on spatiotemporal phagocytic capacity of astrocytes versus microglia is very much appreciated.

Overall, the manuscript has been much improved by these additions, and careful review of the manuscript combined with streamlining of the descriptions of the data contained. The conclusions now more appropriately match the data provided (in both the revised main figures, as well as the extensive supplementary material).

Reviewer #3 (Remarks to the Author):

The authors addressed my comments appropriately, I have no further questions.

Responses to Reviewers comments

We thank all of the reviewers for their careful reading of our manuscript and their many helpful comments and suggestions for revision. We have worked hard to address all of the suggestions and have added many new experiments as well as text changes. A detailed response to each of the referee's comments is given below (referee's comments in black and our responses in blue).

Reviewers' comments:

Reviewer #1 (Remarks to the Author):

This manuscript for Yosuke Morizawa and colleagues aims to address the interesting topic of debris clearance and remodelling following ischemic injury in the cortex of adult mice. They start with a nice descriptive section using immunofluorescent microscopy, showing the different temporal location of phagocytic astrocytes and microglia - suggesting a different role for these two phagocytic cells in the central nervous system following injury. The hypothesis that astrocytes and microglia (both of which are competently able to phagocytose a range of different molecules under different conditions) may play different and non-redundant roles in clearance of damaged CNS cells following injury is intriguing - and provides a target for future research into possible pharmacotherapeutic intervention to help treat such traumatic injuries. Unfortunately, the remainder of the manuscript, though also novel in its approach, fails to address several key flaws in experimental design, which ultimately leave the conclusions of the paper to appear over-reaching and not founded on the data present here within.

Specifically: The main error of this manuscript is the reliance on a cell-culture proxy for in vivo phagocytosis - that is, after showing the engulfment of synapses in vivo (via staining with synaptophysin) the authors migrate to showing engulfment of neuronal debris in vitro, before then using only non-biological fluorescent molecules for the remainder of the study. The only reason given for such an approach is that it is 'easier to quantify' (line 224), which seems a poor proxy. This approach raises many questions:

Do astrocytes only engulf synapses in vivo? What about neuronal debris (their original in vitro test)?

Reply: We apologize that this part was confusing in the original manuscript. We showed that astrocytes engulfed endogenous degenerating neuronal debris labelled by Fluoro-jade B (FJ) as shown in the original version of Fig. 1a-c, e, f (except for original Fig. 1d). Furthermore, astrocytes internalized apoptotic neuronal debris injected exogenously (originally in vitro) in original Fig. 1d. However, the original Fig. 1d caused confusion and was unnecessary; therefore, we have deleted these data in the revised version of the manuscript. Instead, we have added new data showing astrocytes enwrapped NeuN- and FJ-double positive degenerating neurons, and that NeuN⁺ signals colocalized with LAMP2 in astrocytes (revised version Fig. 1b, c). Reviewer #3 also pointed out the necessity of co-detection of a neuronal marker with FJ and astrocytes for the precise demonstration of phagocytosis of neurons. In addition, we have also provided new magnified images of GFAP, Gal-3 and FJ in revised Fig. 1d because higher resolution images were also requested by Reviewer #3. Furthermore, we have added new 3-dimensional electron microscopy (3D-EM) analysis that clearly shows astrocytes contain numerous phagosomes after MCAO (revised Fig. 2 and revised Supplemental Fig. 4.). Please also see below comments for Reviewer #3.

RE-REVIEW: The main concern with the original submission of this manuscript was the in back-and-forth use of debris/synapses in in vitro engulfment proxys, and overtranslation to in vivo importance of ingestion of fluoro jade beads that was not substantiated. The authors have gone some way to rectifying this problem by largely removing the original data. New data showing engulfment of Fluoro-Jade-labelled neuronal debris in vivo goes some way to rectifying this confusion. The addition of higher-resolution images in FIG 1D is appreciated - with the addition of 3-dimensional EM images (Fig 2) this engulfment of these neuronal debris is more comprehensively show as internalised in astrocytes.

What about debris from other dying cells - eg. Oligo lineage and endothelial cells that would be damaged in such an ischemic injury?

As pointed out, astrocytes might engulf other dying cells including oligo lineage and endothelial cells. However, we did not find any evidence that oligodendrocytes and endothelial cells were engulfed by astrocytes when analysed by immunohistochemistry (IHC) combined with confocal microscopy. We used anti-oligodendrocyte (MAB 1580; Millipore) and anti-CD31 antibodies (AB312908 BioLegend) to stain oligodendrocytes and endothelial cells, respectively. However, both of these signals were not observed in GFAP⁺

astrocytes and even in Iba1⁺ microglia 7 days after MCAO (data not shown). Therefore, it is not clear whether astrocytes engulf these types of cells or that the detection of these cells engulfed by astrocytes might be difficult using IHC and confocal microscopy. Judging from the results that these cells were not detected even in microglia after MCAO, the latter would be more probable. Alternatively, antigens of oligodendrocytes or endothelial cells might disappear soon after cell death, damage or when engulfed by glial cells, and thus, IHC is not suitable for their detection. However, we must await further analysis to clarify this.

Does the change in phagocytosis of the non-biological fluorescent molecule actually represent a biologically relevant alteration? Would this change in phagocytosis be seen in vitro if using biologically relevant neuronal debris or synapses? Could these pharmacological blockade experiments be conducted in vivo to see a change in phagocytic capacity?

We should have explained that these fluorescent beads (carboxylate or sulfate-coated latex beads) are commonly used as simplified targets that mimic the negative-surface charge of apoptotic cells in phagocytosis research (Kiss et al., Curr Biol 2006; Erwig et al., PNAS 2006; Park et al., Nature 2007; Koizumi et al., Nature 2007). Indeed, the uptake of these beads was decreased by inhibitors of phagocytosis as for the uptake of apoptotic neuronal debris and synaptic debris (Supplemental Fig. 9g-i). In addition, we have newly added data that ABCA1-deficient astrocytes have less phagocytic ability for neuronal debris engulfment as well as fluorescent beads compared with wild-type astrocytes in the revised version of Supplementary Fig. 10g.

With regard to the use of pharmacological blockers in vivo, it is difficult to use actin polymerization or PI3K inhibitors because they have many non-specific and toxic effects in vivo. An ABCA1 inhibitor, Glyburide, which is an antidiabetic drug, could be used to inhibit the ABCA1-dependent phagocytosis of astrocytes after ischemic injury. However, it is easier to use astrocyte-specific ABCA1 knockout mice to manipulate ABCA1 in astrocytes in vivo, and thus, as shown in revised Fig. 8, we have prepared and used such conditional knockout mice. We have clearly shown the in vivo relevance of ABCA1 for the engulfment of debris by astrocytes after ischemic injury. Similar comments were also raised by Reviewer #3, and we have newly included several in vivo

data in the revised manuscript (please see below, and new Fig. 8, Supplemental Figs. 11-14).

RE-REVIEW: The new addition of experiments using ABCA1^{-/-} mice is nice, and a greater improvement on the suggestion, by this reviewer, to validate in vitro experiments using pharmacological blockade. These new data (FIG 8, and SUPP FIGs 11-14) add a large amount of confidence to the manuscript.

Is the Mertk/Megf10/Abca1 pathway equally important for the engulfment of all targets by astrocytes? Are the differences for synapses, neuronal debris, myelin debris, etc.? Are the differences stated here simple a component of the engulfment of non-biological fluorescent molecules?

We thank you for these interesting remarks. As mentioned in the Discussion, astrocyte-like glial cells eliminate neuronal subcompartments via the context-dependent use of distinct engulfment pathways during larval metamorphosis in Drosophila (reference 65). Although many scientists have already investigated molecules involved in phagocytosis machineries, their specificities for targets, i.e., synapses, neuronal debris, and myelin debris, are poorly understood, and less is known about the mechanisms involved. Both MerTK and MEGF10 function as engulfment receptors by recognizing opsonin proteins bound to phosphatidylserine presented in target debris. MerTK recognizes Gas-6 and Protein S, and MEGF10 recognizes C1q and an unidentified molecule in mammals (Lew et al., eLife 2014; Iram et al., J Neurosci 2016; Wang et al., Nat Cell Biol 2010). Although the exact function of ABCA1 in engulfment is still unknown, it is required for the engulfment function of MEGF10 pathways (references 32-41 in the revised manuscript). As described above, we clearly showed that ABCA1 is important for the engulfment of debris both in vitro and in vivo (revised Figs. 7, 8). In this manuscript, we have focused on the fact that astrocytes become phagocytic after MCAO, and that ABCA1 is a key molecule in the process. Thus, we think that to clarify whether MerTK/MEGF10/ABCA1-pathways discriminate individual targets for their phagocytosis is slightly out of context for the current study and should be part of our next study.

Re-REVIEW: I agree that these experiments, if not already completed are outside the scope of the current manuscript. It is an enticing avenue for further study and I look forward to the future progress from this group. The close association between ABCA1 and MEGF10 will be very interesting to watch as the story unfolds further.

The authors state on several occasions that their astrocytes are able to engulf myelin debris (eg. Fig 1), however no experiment is designed to test this, nor is any data shown to validate these statements. They should be removed.

Myelin structures can be observed under electron microscopy (EM), and were clearly detected in astrocytes after ischemic injury by immuno-EM (Fig. 1e, an arrow indicates myelin-like structure). We have newly added these data using 3D-EM to revised Fig 2h.

RE-REVIEW: this inclusion is much appreciated

Additionally, at line 288, they state that astrocytes are able to engulf immune cells - again with no data to substantiate these claims. Please remove.

We have provided new clearer images and line profile status in revised Supplementary Fig. 3, showing the presence of Iba1⁺ fractions in LAMP2⁺ lysosomes in reactive astrocytes after ischemic injury.

RE-REVIEW: this inclusion is much appreciated, however the statement should read that astrocytes engulf 'fragments' of immune cells, not entire cells as was first suggested. The appropriate term has been used in the title of SUPP FIG 3.

Line 231 - do these astrocytes engulf more, or less, neuronal debris than activated astrocytes?

We think that cultured astrocytes we used are activated astrocytes. Previous transcriptome analysis of astrocytes cultured from neonatal brain by the classical method (McCarthy and de Vellis, 1980) showed that they are very different from mature astrocytes in the healthy brain (Cahoy et al., 2008; Foo et al., 2011). As stated in the manuscript (lines 225-229), these cultured astrocytes have similar genetic features with reactive astrocytes after MCAO in vivo, and thus we think that these astrocytes show similar functions to activated astrocytes, i.e., these astrocytes would engulf neuronal debris as much as the activated astrocytes.

RE-REVIEW: This is an interesting thought. I look forward to seeing any future studies this group completes on the topic.

(line 69) - do these phagocytic astrocytes, once induced, ever return back to a 'non-phagocytic' phenotype? Or once transitioning into this state, do they remain?

Data in Figs. 4 and 5 (original Figs 2, 3) showed that phagocytic marker (Galectin-3 and LAMP2) immunoreactivities in astrocytes peaked at 7 days and then decreased significantly at 14 days after MCAO. This suggests that phagocytic astrocytes gradually return to a non-phagocytic phenotype.

RE-REVIEW: thank you

Similarly, on line 235, do these pharmacological approaches cause a decrease in astrocyte reactivity transcripts? Do they alter other machinery within the cells, or only the phagocytic pathways?

We have not tested whether Glyburide and PSC833 affect astrocyte reactivity transcripts. However, these chemicals did not inhibit engulfment by ABCA1-deficient astrocytes indicating that they might inhibit ABCA1 functions leading to the inhibition of engulfment. We have added these results to Supplemental Fig. 10g.

RE-REVIEW: This may be, as by the authors own admission above, their culture systems contain reactive astrocytes – which presumably do not undergo the changes that one would expect to see in astrocytes in the context of the changing microenvironment in a stroke lesion. Having said this, the MEGF10 decrease in phagocytosis (panel H) is similar to that seen for knockout of MEGF10 in vivo – suggesting that their cultures are responsive to some level of phagocytic alteration.

Line 99 - STAT3 does not label all reactive astrocytes, only one of the (possibly many) activation states of these cells.

This was also raised by Reviewer #3. To avoid confusion, we have deleted these data from the revised version of the manuscript.

Figure 4, and line 268 - Abca1 expression has not been reported as upregulated in astrocytes following ischemic injury (the data provided here in Fig. 4 is of whole brain, with a mix of multiple cell types). How can the authors fit this known literature with their cell culture experiments in which they see an increase in this transporter? How appropriate is their cell culture model for interpretation of an astrocyte following ischemic injury?

We apologize for the confusing text in the original manuscript. We validated Abca1 mRNA and protein upregulation in astrocytes after MCAO using in situ hybridization and IHC staining in the revised Fig. 6 (original Fig. 4). Thus, we think that to increase ABCA1 in cultured astrocytes and to investigate its functional role in engulfment is appropriate. However, which molecules increase ABCA1 in astrocytes after ischemia is unknown. We use the LXR agonist T0901317 to increase ABCA1 in gain of function assay, i.e., whether the increase in ABCA1 enhances astrocytic engulfment. It is well known that LXR agonists induce Abca1 in astrocytes and other cells (Costet et al., JBC 2000; Venkateswaran et al., PNAS 2000; Chawla et al., Mol Cell 2001; Terwel et al., J Neurosci 2011). We did not use the LXR agonist T0901317 to mimic culture models of ischemic injury.

RE-REVIEW: the inclusion of in situ images for Abca1 is much appreciated

In conclusion, the overall thesis of Morizawa et al., is highly interesting, and the pathophysiological role of such activated astrocytes is of great interest to not only the glial biology community, but also the wider neuroscience community as a whole. Unfortunately, the conclusions of this manuscript are not substantiated by the data provided, though if they were I would highly recommend this manuscript for consideration for publication. In its current state however, major revisions would be required to bring the manuscript up to the standards set by this journal.

Additional small comments:

Where there no phagocytic astrocytes during earlier stages of ischemic injury recovery? Similarly, where there no phagocytic microglia during later stages? These data are alluded to, but not specifically stated.

These data were provided in Figs. 4 and 5 (original Figs 2, 3). We used the term 'phagocytic' based on the enhancement of phagocytic markers Galectin-3 and lysosomal protein LAMP2 (astrocytes) or CD68 (microglia) compared with the uninjured side (contra). We just stated there were more phagocytic microglia during earlier stages of ischemic injury and there were more phagocytic astrocytes during later stages based on these data.

RE-REVIEW: thank you for clarification, this was ambiguous in the original manuscript

Line 53 - aside from resident microglia, what about the infiltration of other, peripheral professional phagocytes like macrophages or neutrophils, which are known to move into the CNS following such ischemic injuries (as well as other trauma).

This was stated and modified in Line 96 of the manuscript. As you pointed out, we confirmed that CD45^{high+} and CD11b^{high+} immune cell infiltration from the periphery occurred in our MCAO model (data not shown). However, we could not discriminate microglia from other peripheral immune cells in this study.

Line 117 - I am not sure one can make comments about the process-bearing nature of astrocytes from EM images, without completing serial section reconstructions of cells. Can the authors provide any comment on the complexity of the processes - perhaps from IF or other low power light microscopy?

Thank you for this comment. In the original manuscript, iEM data showed that GFAP-immunopositive astrocytes had phagocytic inclusions. As you stated, it is difficult to discriminate astrocyte fine processes from EM images without IHC staining or a serial sectioning approach. Thus, we have conducted and added 3D-EM data to revised Fig. 2 and Supplemental Fig. 4. In the stack of SBF-SEM (3D-EM) serial images, astrocytes were identified as cells, which were continuous from processes containing bundles of intermediate filaments and often surrounding basement membranes of blood vessels and/or synaptic connections.

RE-REVIEW: These additions are greatly appreciated, and do not leave the interpretation ambiguous.

Minor suggestion (also relevant for some other figures):

arrows/arrowheads in some panels are incredibly small and difficult to see (eg. FIG2 G/H). These should be made larger

Line 135 - a possible discrepancy arises - as above it is stated that galectin-3 positive cells are GFAP positive astros? (as measured as FJ +ve cells?), while here galectin-3 signal is co-localised with Iba1 positive microglia.

As shown in Fig. 4, during the earlier stages of ischemic injury (days 1 and 3 after MCAO), most Galectin-3⁺ cells were Iba1⁺ microglia in the ischemic core

region. During the later stages (days 7 and 14 after MCAO), most Galectin-3⁺ cells were GFAP⁺ astrocytes especially in the penumbra region. Iba1⁺ microglia were also Galectin-3⁺ but only in the ischemic core region at day 7 after MCAO and the immunoreactivity was much lower than at day 3 after MCAO.

RE-REVIEW: thank you for clarification here, and in the text

Line 149 - the marker 3PGDH is not a common reactive astrocyte marker, and has not been introduced well to this section.

We apologize for lack of explanation about 3PGDH. We used 3PGDH as a pan-astrocyte marker not a reactive astrocyte marker because the GFAP antibody did not label astrocytes in the uninjured striatum. A description and reference have been provided in line 171 of the revised version of the manuscript (Furuya et al., PNAS 2000; Yamasaki et al., J Neurosci 2001; Ehmsen et al., J Neurosci 2013).

Re-REVIEW: 3PGDH (Phgdh) in mouse, is expressed by all glial subtypes (astrocytes, oligodendrocytes, and microglia). Did the authors see staining in cells other than astrocytes? Other markers of astrocytes (eg. S100b, Aldh1l1) could be telling of the specificity of this new marker.

Line 154 - how long did the CD68 positive microglia persist following ischemic injury? There are reports in the literature of activated microglia being present several months following traumatic injury in the spinal cord.

Although Iba1⁺ microglia persisted over 2 months in the core following ischemic injury (data not shown), CD68 expression dropped to the level of uninjured striatum by 14 days after MCAO (revised Fig. 5).

Re-REVIEW: thank you for the inclusion of new data

Line 221 - what is the biological relevance of completing a phagocytosis assay at 4°C?

The incubation of cells at a low temperature is a commonly used negative control for ingestion. In such conditions, phagocytes can bind targets, but the internalization of particles is largely prevented (Peterson et al., Infect Immun 1977; Mondal et al., Comp Biochem Physiol A Mol Integr Physiol 2001; Mao et

al., JCB 2009). It is thought that low temperatures reduce membrane fluidity and influence many other pertinent cellular properties or activities.

Line 241 - what are these wild type astrocytes? Earlier in the manuscript the authors state that their cell culture model already produces cells that are like stroke-induced reactive astrocytes.

Thank you for indicating this error. "Wild type astrocytes" has been replaced with "non-treated astrocytes" in the revised version of the manuscript (Line 262).

ABCA1 staining in Fig. 5a appears very high. Is there a baseline level of expression broadly across the cell surface of all astrocytes? - or is this simply non-specific staining of the antibody?

As you stated, ABCA1 signals seem to be expressed broadly throughout astrocytes. In addition, when astrocytes bind to beads, they are localized at the bead binding sites. To analyse their spatial distribution more precisely, we performed additional experiments, and the expression of ABCA1-GFP was forced in cultured astrocytes. ABCA1-GFP signals were localized broadly in the surface membrane and at the bead binding sites (arrowheads) as shown below.

The distribution pattern of ABCA1-GFP was associated with ABCA1 as shown in Fig. 7a, and thus we think that ABCA1 signals are not simply non-specific staining of the antibody.

Re-REVIEW: A very interesting finding, and I thank the authors for this additional information.

Line 327 - do the transcript signatures of the pathophysiological astrocytes resemble developmental/embryonic/immature astrocytes? The literature does not support this comment.

As you stated, the original manuscript was speculative. We have toned down the expression as follows.

Re-REVIEW: Appreciated

Line 383

During brain development, network structures and functions dynamically change as well as after acute brain injury, and the brain becomes more plastic than normal adult brain under these situations. Accumulating evidence indicates both astrocytes and microglia play roles in remodelling during these dynamical plastic changes^{4, 56, 57, 58, 59, 60, 61, 62}. Therefore, there might be functional and genetic similarities, under pathophysiological conditions, adult astrocytes might transform into a more phagocytic state similar to immature and/or developmental astrocytes. Together, these findings suggest that astrocytes can phagocytize adjacent materials, including synapses, although this occurs primarily during development or under pathophysiological conditions in the adult brain.

Line 336 - which Dock?

Line 336 - which Elmo?

We apologize for the inadequate explanation. It is known that Crk II, Dock180 and Elmo1 mediate phagocytosis (Albert et al., Nat Cell Biol 2000; Gumienny et al., Cell 2001; Park et al., Nature 2007; Elliott et al., Nature 2010; Lu et al., Nat Cell Biol 2011). We have added these references and statements to the revised version of the manuscript (Line 407).

Re-REVIEW: Appreciated

Line 434 - two rectal thermometers?

We thank you for noticing the duplication. We have deleted the redundant sentence in the revised manuscript.

Fig. 1 - state what the gold particles in immune-EM are labelling. Is it synaptophysin?

In Fig. 1, gold particles were used to label the GFAP antibody. Although we described this in Figure 1g and the legend in the original version of the manuscript, we have described this in the text body of the revised manuscript (Line 113).

Fig. 4 - continuity, place boxes around all or no panels.

We thank you for noticing this error. We have corrected this mistake in revised Fig. 6 (original Fig. 4).

In general protein is not 'expressed', mRNA is expressed. This should be fixed throughout the manuscript.

Multiple continuity problems exist with gene and protein names. Ideally, genes should be italicised with a capitalised first letter (eg. *Gfap*, *Mertk*), while protein names should be capitalized (eg. GFAP, MERTK). This manuscript switches on several occasions - eg. *Mertk*, MERTK, MerTK.

We thank you for the comment regarding the incorrect terminology and abbreviations. We have deleted "expression" when describing proteins, and have corrected all gene and protein names in the revised version of the manuscript.

Re-REVIEW: Appreciated. The continuity makes understanding the complexity of the manuscript much easier for the reader.

There are several grammatical errors throughout the manuscript (eg. Line 50 '...thousands of debris...', line 196 'Although the another...') These should be carefully edited out of future versions of the manuscript.

We thank the reviewer for this comment. The revised manuscript has been edited and proofread by a native-English speaker.

Reviewer #2 (Remarks to the Author):

This manuscript reports several lines of evidence that in penumbra regions around forebrain stroke, certain reactive astrocytes become phagocytic and engulf debris from neurons, myelin and other cell types. The evidence presented includes identification of engulfed fluorescently labeled debris within immunoreactive astrocytes, also co-localized with the lysosomal marker

LAMP2, and the phagocytosis associated molecule Galectin-3. In addition, immune-electron microscopy was used to provide ultrastructural evidence for engulfment of debris by astrocytes. Astrocyte engulfment of debris was found to peak around 7 days after stroke, but began around 3 days after stroke and continued for at least 14 days. Genetic regulation of astrocyte phagocytic activity was associated with ABCA1 pathway, the structural homolog of the *c. elegans* engulfment regulating gene, *ced-7*. Various loss- or gain-of-function experiments implicated the ABCA1 pathway in regulating astrocyte phagocytic activity.

The various experiments appear to have been well conducted and appropriately controlled. The data processing and statistical analyses seem appropriate and rigorous. The figures are of good quality. I found that the data convincingly support the interpretations presented in the text. The discussion is balanced and the text is well written.

Although evidence for phagocytic roles for astrocytes during development, in particular with regard to synapse pruning, at present there is little information available on potential roles of reactive astrocytes in phagocytic activities of debris or of synapses that might be dysfunctional. This study provides strong evidence that reactive astrocytes take part in such phagocytic activities in penumbra regions after stroke. The findings are likely to be of interest in the stroke field, and more broadly in other contexts of CNS damage or degeneration that might generate debris in need of phagocytosis and degradation. I have no major criticisms, but have one concern that requires attention.

Specific comments and concerns:

1. The first sentences of the Abstract and Discussion both state that "Adult astrocytes are typically quiescent...", but this is not correct, and it is not clear why the authors would want to make such a comment. Adult astrocytes are not quiescent but are very active in healthy tissue, where they exert numerous critical functions in response to many dynamic signaling events. The notion that astrocytes are "quiescent" in healthy tissue is antiquated and should not

be perpetuated. This statement should be removed or edited in both the Abstract and Discussion. For example, "In addition to their functions in healthy tissue, astrocytes become reactive in response to various brain insults." There are many good reviews on dynamic astrocyte activity in healthy tissue that the authors may wish to look at.

We thank you for these kind comments and pointing out the inappropriate description of healthy astrocytes in the adult brain. In the revised manuscript, we have now replaced "Adult astrocytes are typically quiescent..." with "Astrocytes become reactive following various brain insults; however, the functions of reactive astrocytes are poorly understood." in the Abstract and deleted the sentence in the Discussion.

Reviewer #3 (Remarks to the Author):

In the research article by Morizawa et al., the authors present complex in vitro studies and in vivo histological work demonstrating the role of astrocytes as phagocytic cells in cell cultures and in the brain after experimental cerebral ischemia. They also show that phagocytosis by astrocytes is partially mediated by the ABCA1 pathway. In general, the paper is interesting and the observations made could advance the understanding of the field. There are, however, some major issues which reduce the impact of the findings and should be considered by the authors.

The fact that astrocytes can phagocytose cell debris after ischemia is interesting. Phagocytosis by astrocytes has been reported previously, however the present findings are the first to show this in the ischemic brain. One major issue is that the functional role of phagocytic activity of astrocytes in neuronal loss, clearance of synapses, remodelling or overall outcome after cerebral ischemia has not been investigated in the paper. In other words, it remains unclear how much debris, synaptic structures, etc., is phagocytosed by astrocytes in the brain compared with the phagocytic activity of microglia, macrophages or other leukocytes that are abundant in the brain 3-14 days after MCAo and it is also unclear whether blockade of astrocyte-mediated phagocytosis would change the injured brain milieu in any way in the present experimental model.

We agree with your comments and understanding the functional aspects of phagocytosis by astrocytes is of great interest to us. We have performed additional in vivo experiments to address these issues in the revised version of the manuscript.

- 1. Reactive astrocytes enwrapped many FJ⁺ small fractions (Fig. 1d), but the enwrapment of FJ⁺ large neuronal cell bodies by reactive astrocytes was less frequent compared with microglia (Fig. 3). However most processes were very close to the large debris.*
- 2. 3D-EM analysis clearly showed that reactive astrocytes frequently contained small debris in their cytoplasm (20/22 observed processes had engulfed debris, Figs. 2, 8, Supplemental Fig. 4).*

3. 3D-EM analysis showed that 30% of observed microglia had large debris (maximum Feret diameter > 4 μm, Supplemental Fig. 14b). However, the processes of observed astrocytes contained no large debris, although several processes were attached to large debris (Fig. 2g, 8a). Astrocytes appear to engulf mainly small cellular debris or possibly may break large debris into smaller ones and then engulf them.

4. Engulfed small debris densities in microglia and astrocytes were comparable, indicating that astrocytic phagocytosis was as frequent as microglia phagocytosis (Fig. 8g).

5. To confirm the importance of ABCA1 in astrocytic phagocytosis in vivo, we generated astrocyte-specific ABCA1 knock out (ABCA1 cKO) mice, and demonstrated that ABCA1-deficient astrocytes in vivo had less phagocytic ability after MCAO (Fig. 8a-d, Supplemental Figs. 13, 14, please also see a below comment in detail).

6. Consistent with a deficit in engulfment by astrocytes from ABCA1 cKO mice, extracellular debris tended to be higher in ABCA1 cKO tissue, although the difference was not significant because of the large variance (Fig. 8e).

Because astrocyte processes are very complex and too thin to quantify precisely, and a comparison of the total amount of engulfed debris using light microscopy might overestimate or underestimate values, we conducted 3D-EM analysis to compare and show how much debris was engulfed by astrocytes and microglia (new Fig 8).

With regard to the outcome of astrocytic phagocytosis, we are now investigating the functional roles and consequences by histological and behavioural long term approaches using astrocyte-specific ABCA1 cKO mice, whose astrocytes are less phagocytic after MCAO. However, we have not reached any conclusion yet, and it will take a long time to finalize these results. Thus, we think that these issues are likely to be solved in our next study.

It would also be important to get an idea about the kinetics of astrocyte phagocytosis in vivo as in the paper only histological data suggest the association of cell debris with astrocytes, which are mostly outside the cells. Have the authors tried to follow these processes with live confocal or two-photon imaging? This could greatly increase the impact of the paper, as

phagocytic activity is mostly devoted to microglia in the injured brain, which has been captured several times before using in vivo imaging.

We agree that imaging of astrocytic phagocytosis would make our study more impressive and convincing. We have tried to perform these experiments in acute slices with two-photon microscopy, but so far we have not succeeded in imaging of astrocytic phagocytosis because of technical difficulties. We are now actively working on this with new genetically encoded fluorescent probes to visualize the fine processes of astrocytes and degenerating neurons. It will take a long time to achieve these results, and we feel it would be the basis of an independent paper. Thus, we would like to leave for our next study.

The upregulation of ABCA1 in response to cerebral ischemia in astrocytes seems to be a convincing observation. However, while using multiple approaches to demonstrate that ABCA1 contributes to phagocytic activity of astrocytes in vitro, the authors did not make efforts to investigate this functionally in vivo. It could be done by either using KO mice or pharmacological approaches in order to demonstrate the relevance of these mechanisms in the injured brain. *We agree that astrocytic phagocytosis should be validated in vivo. Therefore, we performed additional in vivo experiments by 3D-EM analysis and immunohistochemical (IHC) staining. As shown above, we compared the phagocytic activity of astrocytes in *Abca1^{flox/flox}:: Cre*-negative mice (control) and *Abca1^{flox/flox}:: GFAP-Cre* mice (astrocyte-specific ABCA1 conditional knockout: cKO) (Supplemental Figs. 11, 12). Using 3D-EM analysis, we found that astrocytic processes in the ipsilateral striatum of WT mice engulfed a large amount of debris after MCAO (Fig. 2, Supplemental Fig. 4), whereas those in ABCA1 cKO mice engulfed less debris (Fig. 8a-b). In addition, using IHC analysis, we also found that reactive astrocytes in ABCA1 cKO mice had significantly less LAMP2 and Galectin-3 signals compared to control mice after MCAO (Supplemental Fig. 13). We have added these results to new Fig. 2, Supplemental Fig. 4, Fig. 8a-b, Supplemental Fig.13 and the corresponding text in the revised version of the manuscript.*

Further points:

- Confocal microscopic analysis represents a key approach in the paper, but no information is provided regarding the number of Z planes recorded, the step size used and how subsequent analysis was performed. This should be described in detail. In many figures the authors show relatively low resolution images of very small particles to demonstrate phagocytic activity of astrocytes (e.g. Fig1a-f).

Supplementary Fig. 3d and 3e are also good examples where colocalisation of Iba1-positive particles with LAMP2 is very difficult to interpret and no quantitative data is available about incidence of colocalisation between LAMP2 and Iba1. This should be provided.

Thank for pointing out these inadequacies. We have added information about the Z planes and step size used in all revised figure legends. We have also added an explanation of further analyses performed in the "Image analysis" section of the Methods in the revised manuscript. We apologize for the low-resolution images. Our figure resolutions were originally not low, and therefore we have changed the method of uploading and provided greater magnified images in the revised manuscript.

With regard to Supplemental Fig. 3, we recaptured and reanalysed the colocalisation of Iba1⁺ particles with LAMP2 in astrocytes and have added quantitative data showing colocalisation between Iba1 and LAMP2 signals in GFAP⁺ astrocytes (Supplemental Fig. 3) in the revised manuscript.

- STAT3-mediated signal transduction is present in multiple cell types (neurons, microglia, macrophages, astrocytes), therefore STAT3 should not be used as a reactive astrocyte marker. It however could indicate glial - neuronal activation in response to injury. The authors should quantify STAT3 expression in different cell types in the brain after MCAo and discuss the results in the paper accordingly.

This was also raised by Reviewer #1, and we have deleted the data showing increased STAT3 in astrocytes because this was confusing and unnecessary for our point in the current study.

- It is not sufficient to show Fluoro-jade B (FJ) staining in astrocytes (Fig.1) whilst stating that they phagocytose neuronal debris after cerebral

ischemia. FJ can also stain degenerating astrocytes and other cells (e.g. Damjanac et al., 2007; Anderson et al., 2003). The authors must perform co-detection of a neuronal marker with FJ and astrocytes at the same time to convincingly demonstrate phagocytosis of neurons and show quantitative data on how frequent astrocytic phagocytosis is compared to that of microglia. In addition, it has to be confirmed whether phagocytosis by astrocytes takes place independently of microglial phagocytosis i.e. FJ, Iba1 and GFAP staining should also be shown. This latter combination is also crucial since the vast majority of FJ staining is seen outside of astrocytes as shown in Fig.1b and it is similar in the experiment when fluorescently labelled apoptotic neurons were injected in the brain (Fig.1d). Showing galectin-3 staining in Fig.2. is not sufficient to represent the actual phagocytic activity of astrocytes and microglia. Higher resolution confocal images should also be shown for Figs.1-3.

Thank you for these important comments. We have conducted IHC with FJ staining and shown that astrocyte processes enwrap NeuN- and FJ-double positive signals (Fig. 1b). Also, NeuN⁺ small fractions in astrocytes colocalized with LAMP2 signals in astrocytes (Fig. 1c).

We knew the previous studies showing that FJ could stain reactive astrocytes and microglia, and therefore, we carefully conducted FJ staining so as not to stain reactive astrocytes by using a lower FJ concentration and shorter reaction time. We have added these points to the Methods section in the revised version of the manuscript as follows;

Line 574

The reaction time and concentration of FJ solution were also decreased from 30 min and 0.001% to 10 min and 0.0001%, to avoid non-specific staining.

Using the procedure above, we clarified that GFAP⁺ reactive astrocytes did not have FJ-positive patterns, as seen in degenerating neurons and debris (Fig. 1, 3). Furthermore, our analysis showed that 100% of FJ⁺ large signals were colocalized with NeuN⁺ neuronal signals (Supplemental Fig. 1d). We have added these results to the revised version of the manuscript (Line 91).

In revised Fig. 3, we have shown the frequency of astrocyte processes enwrapping FJ⁺ large debris compared with microglia by IHC analysis. However, these analyses along with confocal microscopy and cellular markers

is limited for detailed studies. Thus, we performed 3D-EM studies and compared the phagocytic abilities of astrocytes and microglia. Using this method, we found that astrocytic engulfment of debris was as frequent as microglia engulfment (small debris; maximum Feret diameter < 4 μm). We have added these results to revised Fig. 8g.

IHC staining of Iba1 and GFAP with FJ staining were shown in Fig. 3a. Based on these data, we found that astrocytic phagocytosis occurred, at least in part, independently of microglial phagocytosis: (1) GFAP⁺ astrocytic processes enwrapped FJ⁺ large signals independently of microglia; (2) GFAP⁺ processes and Iba1⁺ processes enwrapped FJ⁺ large signals exclusively, and (3) GFAP⁺ processes enwrapped Iba1⁺ processes including FJ⁺ large signals. However, we need to clarify these points in more detail in the future.

- Supplementary Fig. 3: "We also found that Iba1+ signals localized with LAMP2+ lysosomal vesicles in the reactive astrocytes, indicating that phagocytic astrocytes engulf neuronal debris as well as immune cell debris." It is an interesting observation, but the majority of the Iba1 staining appears outside the astrocyte. The above statements have to be supported with higher resolution images and quantification, the present resolution and level of detail in the images is not sufficient to conclude.

We agree with this comment. As stated above, we conducted IHC analysis and have now provided high quality images showing many Iba1⁺ fractions were within GFAP⁺ astrocytes, some of which were colocalized with LAMP2. We have added these data and quantitative analysis to the revised version of the manuscript (Supplemental Fig. 3).

- Fig.5a: "Arrowheads indicate ABCA1 accumulation and points of attachment between captured beads and astrocytes." There are no arrowheads in the panels.

Thank you for pointing out this error. We have added arrowheads to revised Figure 7a.

- Details about how the authors assessed astrocyte phagocytosis by FACS should be shown in Supplementary Fig. 8. The authors should give examples to how they discriminated unbound cells or beads from astrocytes

using forward and side-scatter and also explain this better in the methods. How did the authors assess the phagocytosis of synaptosomes *in vitro*? Did they prepare synaptosome fractions or just stained for synaptophysin after addition of PKH26 cells? If the latter, the amount of synaptophysin staining in astrocytes is difficult to interpret.

We apologize for the lack of explanation and reference regarding the in vitro phagocytosis assay and FACS analysis. As shown below, we easily distinguished apoptotic neuronal debris and fluorescent beads from cultured astrocytes based on their homogeneity and small size using forward (FSC; horizontal) and side-scatter plots (SSC; vertical). Signals inside the gates are detached targets; therefore, we analysed signals outside the exclusion gate.

With regard to synaptosome engulfment, we prepared a synaptosomal fraction from mouse brain by multiple centrifugations and labelled cells with PKH26. Similar to neuronal debris, we incubated these PKH26⁺ synaptosomal

fractions with cultured astrocytes and quantified their phagocytic ability by FACS. To confirm that PKH26⁺ signals were synaptic debris, we performed synaptophysin 1-staining (Supplemental Fig. 9f). We have added explanations to the revised version of the Methods (Line 707).

Re-Responses to Reviewers comments

We thank the reviewer for their careful reading of our manuscript and their helpful comments and suggestions for revision. A detailed response to the referee's comments is given below as Re-reply (our responses in green).

Reviewers' comments:

Reviewer #1 (Remarks to the Author):

This manuscript for Yosuke Morizawa and colleagues aims to address the interesting topic of debris clearance and remodelling following ischemic injury in the cortex of adult mice. They start with a nice descriptive section using immunofluorescent microscopy, showing the different temporal location of phagocytic astrocytes and microglia - suggesting a different role for these two phagocytic cells in the central nervous system following injury. The hypothesis that astrocytes and microglia (both of which are competently able to phagocytose a range of different molecules under different conditions) may play different and non-redundant roles in clearance of damaged CNS cells following injury is intriguing - and provides a target for future research into possible pharmacotherapeutic intervention to help treat such traumatic injuries. Unfortunately, the remainder of the manuscript, though also novel in its approach, fails to address several key flaws in experimental design, which ultimately leave the conclusions of the paper to appear over-reaching and not founded on the data present here within. Specifically: The main error of this manuscript is the reliance on a cell-culture proxy for in vivo phagocytosis - that is, after showing the engulfment of synapses in vivo (via staining with synaptophysin) the authors migrate to showing engulfment of neuronal debris in vitro, before then using only nonbiological fluorescent molecules for the remainder of the study. The only reason given for such an approach is that it is 'easier to quantify' (line 224), which seems a poor proxy. This approach raises many questions: Do astrocytes only engulf synapses in vivo? What about neuronal debris (their original in vitro test)?

Reply: We apologize that this part was confusing in the original manuscript. We showed that astrocytes engulfed endogenous degenerating neuronal debris labelled by Fluoro-jade B (FJ) as shown in the original version of Fig. 1a-c, e, f

(except for original Fig. 1d). Furthermore, astrocytes internalized apoptotic neuronal debris injected exogenously (originally in vitro) in original Fig. 1d. However, the original Fig. 1d caused confusion and was unnecessary; therefore, we have deleted these data in the revised version of the manuscript. Instead, we have added new data showing astrocytes enwrapped NeuN- and FJ-double positive degenerating neurons, and that NeuN+ signals colocalized with LAMP2 in astrocytes (revised version Fig. 1b, c). Reviewer #3 also pointed out the necessity of co-detection of a neuronal marker with FJ and astrocytes for the precise demonstration of phagocytosis of neurons. In addition, we have also provided new magnified images of GFAP, Gal-3 and FJ in revised Fig. 1d because higher resolution images were also requested by Reviewer #3. Furthermore, we have added new 3-dimensional electron microscopy (3D-EM) analysis that clearly shows astrocytes contain numerous phagosomes after MCAO (revised Fig. 2 and revised Supplemental Fig. 4.). Please also see below comments for Reviewer #3.

RE-REVIEW: The main concern with the original submission of this manuscript was the in back-and-forth use of debris/synapses in in vitro engulfment proxys, and overtranslation to in vivo importance of ingestion of fluoro jade beads that was not substantiated. The authors have gone some way to rectifying this problem by largely removing the original data. New data showing engulfment of Fluoro-Jade-labelled neuronal debris in vivo goes some way to rectifying this confusion. The addition of higher-resolution images in FIG 1D is appreciated - with the addition of 3-dimensional EM images (Fig 2) this engulfment of these neuronal debris is more comprehensively show as internalised in astrocytes.

What about debris from other dying cells - eg. Oligo lineage and endothelial cells that would be damaged in such an ischemic injury?

As pointed out, astrocytes might engulf other dying cells including oligo lineage and endothelial cells. However, we did not find any evidence that oligodendrocytes and endothelial cells were engulfed by astrocytes when analysed by immunohistochemistry (IHC) combined with confocal microscopy. We used anti-oligodendrocyte (MAB 1580; Millipore) and anti-CD31 antibodies (AB312908 BioLegend) to stain oligodendrocytes and endothelial cells, respectively. However, both of these signals were not observed in GFAP+ astrocytes and even

in Iba1+ microglia 7 days after MCAO (data not shown). Therefore, it is not clear whether astrocytes engulf these types of cells or that the detection of these cells engulfed by astrocytes might be difficult using IHC and confocal microscopy. Judging from the results that these cells were not detected even in microglia after MCAO, the latter would be more probable. Alternatively, antigens of oligodendrocytes or endothelial cells might disappear soon after cell death, damage or when engulfed by glial cells, and thus, IHC is not suitable for their detection. However, we must await further analysis to clarify this.

Does the change in phagocytosis of the non-biological fluorescent molecule actually represent a biologically relevant alteration? Would this change in phagocytosis be seen in vitro if using biologically relevant neuronal debris or synapses? Could these pharmacological blockade experiments be conducted in vivo to see a change in phagocytic capacity?

We should have explained that these fluorescent beads (carboxylate or sulfate-coated latex beads) are commonly used as simplified targets that mimic the negative-surface charge of apoptotic cells in phagocytosis research (Kiss et al., Curr Biol 2006; Erwig et al., PNAS 2006; Park et al., Nature 2007; Koizumi et al., Nature 2007). Indeed, the uptake of these beads was decreased by inhibitors of phagocytosis as for the uptake of apoptotic neuronal debris and synaptic debris (Supplemental Fig. 9g-i). In addition, we have newly added data that ABCA1-deficient astrocytes have less phagocytic ability for neuronal debris engulfment as well as fluorescent beads compared with wild-type astrocytes in the revised version of Supplementary Fig. 10g. With regard to the use of pharmacological blockers in vivo, it is difficult to use actin polymerization or PI3K inhibitors because they have many non-specific and toxic effects in vivo. An ABCA1 inhibitor, Glyburide, which is an antidiabetic drug, could be used to inhibit the ABCA1-dependent phagocytosis of astrocytes after ischemic injury. However, it is easier to use astrocyte-specific ABCA1 knockout mice to manipulate ABCA1 in astrocytes in vivo, and thus, as shown in revised Fig. 8, we have prepared and used such conditional knockout mice. We have clearly shown the in vivo relevance of ABCA1 for the engulfment of debris by astrocytes after ischemic injury. Similar comments were also raised by

Reviewer #3, and we have newly included several in vivo data in the revised manuscript (please see below, and new Fig. 8, Supplemental Figs. 11-14).

RE-REVIEW: The new addition of experiments using ABCA1^{-/-} mice is nice, and a greater improvement on the suggestion, by this reviewer, to validate in vitro experiments using pharmacological blockade. These new data (FIG 8, and SUPP FIGs 11-14) add a large amount of confidence to the manuscript.

Is the Mertk/Megf10/Abca1 pathway equally important for the engulfment of all targets by astrocytes? Are the differences for synapses, neuronal debris, myelin debris, etc.? Are the differences stated here simply a component of the engulfment of non-biological fluorescent molecules?

We thank you for these interesting remarks. As mentioned in the Discussion, astrocyte-like glial cells eliminate neuronal subcompartments via the context dependent use of distinct engulfment pathways during larval metamorphosis in Drosophila (reference 65). Although many scientists have already investigated molecules involved in phagocytosis machineries, their specificities for targets, i.e., synapses, neuronal debris, and myelin debris, are poorly understood, and less is known about the mechanisms involved. Both MerTK and MEGF10 function as engulfment receptors by recognizing opsonin proteins bound to phosphatidylserine presented in target debris. MerTK recognizes Gas-6 and Protein S, and MEGF10 recognizes C1q and an unidentified molecule in mammals (Lew et al., eLife 2014; Iram et al., J Neurosci 2016; Wang et al., Nat Cell Biol 2010). Although the exact function of ABCA1 in engulfment is still unknown, it is required for the engulfment function of MEGF10 pathways (references 32-41 in the revised manuscript). As described above, we clearly showed that ABCA1 is important for the engulfment of debris both in vitro and in vivo (revised Figs. 7, 8). In this manuscript, we have focused on the fact that astrocytes become phagocytic after MCAO, and that ABCA1 is a key molecule in the process. Thus, we think that to clarify whether MerTK/MEGF10/ABCA1 pathways discriminate individual targets for their phagocytosis is slightly out of context for the current study and should be part of our next study.

Re-REVIEW: I agree that these experiments, if not already completed are outside the scope of the current manuscript. It is an enticing avenue for further study and I

look forward to the future progress from this group. The close association between ABCA1 and MEGF10 will be very interesting to watch as the story unfolds further.

The authors state on several occasions that their astrocytes are able to engulf myelin debris (eg. Fig 1), however no experiment is designed to test this, nor is any data shown to validate these statements. They should be removed. *Myelin structures can be observed under electron microscopy (EM), and were clearly detected in astrocytes after ischemic injury by immuno-EM (Fig. 1e, an arrow indicates myelin-like structure). We have newly added these data using 3D-EM to revised Fig 2h.*

RE-REVIEW: this inclusion is much appreciated

Additionally, at line 288, they state that astrocytes are able to engulf immune cells - again with no data to substantiate these claims. Please remove.

We have provided new clearer images and line profile status in revised Supplementary Fig. 3, showing the presence of Iba1+ fractions in LAMP2+ lysosomes in reactive astrocytes after ischemic injury.

RE-REVIEW: this inclusion is much appreciated, however the statement should read that astrocytes engulf 'fragments' of immune cells, not entire cells as was first suggested. The appropriate term has been used in the title of SUPP FIG 3.

Line 231 - do these astrocytes engulf more, or less, neuronal debris than activated astrocytes?

We think that cultured astrocytes we used are activated astrocytes. Previous transcriptome analysis of astrocytes cultured from neonatal brain by the classical method (McCarthy and de Vellis, 1980) showed that they are very different from mature astrocytes in the healthy brain (Cahoy et al., 2008; Foo et al., 2011). As stated in the manuscript (lines 225-229), these cultured astrocytes have similar genetic features with reactive astrocytes after MCAO in vivo, and thus we think that these astrocytes show similar functions to activated astrocytes, i.e., these astrocytes would engulf neuronal debris as much as the activated astrocytes.

RE-REVIEW: This is an interesting thought. I look forward to seeing any future studies this group completes on the topic.

(line 69) - do these phagocytic astrocytes, once induced, ever return back to a 'non-phagocytic' phenotype? Or once transitioning into this state, do they remain?

Data in Figs. 4 and 5 (original Figs 2, 3) showed that phagocytic marker (Galectin-3 and LAMP2) immunoreactivities in astrocytes peaked at 7 days and then decreased significantly at 14 days after MCAO. This suggests that phagocytic astrocytes gradually return to a non-phagocytic phenotype.

RE-REVIEW: thank you

Similarly, on line 235, do these pharmacological approaches cause a decrease in astrocyte reactivity transcripts? Do they alter other machinery within the cells, or only the phagocytic pathways?

We have not tested whether Glyburide and PSC833 affect astrocyte reactivity transcripts. However, these chemicals did not inhibit engulfment by ABCA1-deficient astrocytes indicating that they might inhibit ABCA1 functions leading to the inhibition of engulfment. We have added these results to Supplemental Fig. 10g.

RE-REVIEW: This may be, as by the authors own admission above, their culture systems contain reactive astrocytes – which presumably do not undergo the changes that one would expect to see in astrocytes in the context of the changing microenvironment in a stroke lesion. Having said this, the MEGF10 decrease in phagocytosis (panel H) is similar to that seen for knockout of MEGF10 in vivo – suggesting that their cultures are responsive to some level of phagocytic alteration.

Re-Reply: As we commented above (in blue fonts), the effects of Glyburide and PSC833 were not observed in ABCA1-deficient astrocytes, suggesting that their main target should be ABCA1 and/or its related pathways. In addition, we have newly demonstrated that ABCA1 is a responsible molecule for the astrocytic phagocytosis in vitro and in vivo in the revised version of the manuscript. Thus, we think that our conclusion is not affected, even if Glyburide and PSC833 may somehow inhibit other phagocytic pathways.

Line 99 - STAT3 does not label all reactive astrocytes, only one of the (possibly many) activation states of these cells.

This was also raised by Reviewer #3. To avoid confusion, we have deleted these data from the revised version of the manuscript.

Figure 4, and line 268 - Abca1 expression has not been reported as upregulated in astrocytes following ischemic injury (the data provided here in Fig. 4 is of whole brain, with a mix of multiple cell types). How can the authors fit this known literature with their cell culture experiments in which they see an increase in this transporter? How appropriate is their cell culture model for interpretation of an astrocyte following ischemic injury?

We apologize for the confusing text in the original manuscript. We validated Abca1 mRNA and protein upregulation in astrocytes after MCAO using in situ hybridization and IHC staining in the revised Fig. 6 (original Fig. 4). Thus, we think that to increase ABCA1 in cultured astrocytes and to investigate its functional role in engulfment is appropriate. However, which molecules increase ABCA1 in astrocytes after ischemia is unknown. We use the LXR agonist T0901317 to increase ABCA1 in gain of function assay, i.e., whether the increase in ABCA1 enhances astrocytic engulfment. It is well known that LXR agonists induce Abca1 in astrocytes and other cells (Costet et al., JBC 2000; Venkateswaran et al., PNAS 2000; Chawla et al., Mol Cell 2001; Terwel et al., J Neurosci 2011). We did not use the LXR agonist T0901317 to mimic culture models of ischemic injury.

RE-REVIEW: the inclusion of in situ images for Abca1 is much appreciated

In conclusion, the overall thesis of Morizawa et al., is highly interesting, and the pathophysiological role of such activated astrocytes is of great interest to not only the glial biology community, but also the wider neuroscience community as a whole. Unfortunately, the conclusions of this manuscript are not substantiated by the data provided, though if they were I would highly recommend this manuscript for consideration for publication. In its current state however, major revisions would be required to bring the manuscript up to the standards set by this journal.

Additional small comments:

Where there no phagocytic astrocytes during earlier stages of ischemic injury recovery? Similarly, where there no phagocytic microglia during later stages? These data are alluded to, but not specifically stated.

These data were provided in Figs. 4 and 5 (original Figs 2, 3). We used the term 'phagocytic' based on the enhancement of phagocytic markers Galectin-3 and lysosomal protein LAMP2 (astrocytes) or CD68 (microglia) compared with the uninjured side (contra). We just stated there were more phagocytic microglia during earlier stages of ischemic injury and there were more phagocytic astrocytes during later stages based on these data.

RE-REVIEW: thank you for clarification, this was ambiguous in the original manuscript

Line 53 - aside from resident microglia, what about the infiltration of other, peripheral professional phagocytes like macrophages or neutrophils, which are known to move into the CNS following such ischemic injuries (as well as other trauma).

This was stated and modified in Line 96 of the manuscript. As you pointed out, we confirmed that CD45^{high+} and CD11b^{high+} immune cell infiltration from the periphery occurred in our MCAO model (data not shown). However, we could not discriminate microglia from other peripheral immune cells in this study.

Line 117 - I am not sure one can make comments about the process-bearing nature of astrocytes from EM images, without completing serial section reconstructions of cells. Can the authors provide any comment on the complexity of the processes - perhaps from IF or other low power light microscopy?

Thank you for this comment. In the original manuscript, iEM data showed that GFAP-immunopositive astrocytes had phagocytic inclusions. As you stated, it is difficult to discriminate astrocyte fine processes from EM images without IHC staining or a serial sectioning approach. Thus, we have conducted and added 3D-EM data to revised Fig. 2 and Supplemental Fig. 4. In the stack of SBF-SEM (3D-EM) serial images, astrocytes were identified as cells, which were continuous from processes containing bundles of intermediate filaments and often surrounding basement membranes of blood vessels and/or synaptic connections.

RE-REVIEW: These additions are greatly appreciated, and do not leave the interpretation ambiguous.

Minor suggestion (also relevant for some other figures):

arrows/arrowheads in some panels are incredibly small and difficult to see (eg. FIG2 G/H). These should be made larger

Re-Reply: Thank you for your suggestion, we replaced them with bigger ones.

Line 135 - a possible discrepancy arises - as above it is stated that galectin-3 positive cells are GFAP positive astros? (as measured as FJ +ve cells?), while here galectin-3 signal is co-localised with Iba1 positive microglia.

As shown in Fig. 4, during the earlier stages of ischemic injury (days 1 and 3 after MCAO), most Galectin-3+ cells were Iba1+ microglia in the ischemic core region. During the later stages (days 7 and 14 after MCAO), most Galectin-3+ cells were GFAP+ astrocytes especially in the penumbra region. Iba1+ microglia were also Galectin-3+ but only in the ischemic core region at day 7 after MCAO and the immunoreactivity was much lower than at day 3 after MCAO.

RE-REVIEW: thank you for clarification here, and in the text

Line 149 - the marker 3PGDH is not a common reactive astrocyte marker, and has not been introduced well to this section.

We apologize for lack of explanation about 3PGDH. We used 3PGDH as a pan-astrocyte marker not a reactive astrocyte marker because the GFAP antibody did not label astrocytes in the uninjured striatum. A description and reference have been provided in line 171 of the revised version of the manuscript (Furuya et al., PNAS 2000; Yamasaki et al., J Neurosci 2001; Ehmsen et al., J Neurosci 2013).

Re-REVIEW: 3PGDH (Phgdh) in mouse, is expressed by all glial subtypes (astrocytes, oligodendrocytes, and microglia). Did the authors see staining in cells other than astrocytes? Other markers of astrocytes (eg. S100b, Aldh111) could be telling of the specificity of this new marker.

Re-Reply: Thank you for your comment. Although 3Pgdh mRNAs might be expressed in other cell types (Barres Brain RNA-seq), we could not find any evidence that 3PGDH protein is expressed by oligodendrocytes and microglia in adult mice brain. Only the literature we could find is a J Neurosci Res paper by

Sugishita et al. (J Neurosci Res. 2001), where they mentioned microglial expression of 3PGDH in rat brain. They could detect 3PGDH in microglia in culture, but their expression level was very low and faint. Furthermore, they described that intense 3PGDH-positive signals were observed in only astrocytes. In addition, a double immunofluorescent study by Yamasaki et al., showed that MRF-1, a microglia-specific Ca²⁺-binding protein, showed no significant overlap with 3PGDH (Fig 4 in Yamasaki et al., J Neurosci, 2001). Moreover, they described that "Even with careful observation, however, we were not able to detect 3PGDH in oligodendrocytic elements, including compact myelin, outer and inner mesaxons, and paranodal cytoplasmic loops (Fig 6 in Yamasaki et al., J Neurosci, 2001)". As shown below, 3PGDH expression seems to be mainly astrocytes-specific staining pattern. Judging from all above findings, we think that 3PGDH is mainly expressed by astrocytes, and that the 3PGDH-positive signals in our immunofluorescent data should be derived from astrocytes. .

Line 154 - how long did the CD68 positive microglia persist following ischemic injury? There are reports in the literature of activated microglia being present several months following traumatic injury in the spinal cord.

Although Iba1+ microglia persisted over 2 months in the core following ischemic injury (data not shown), CD68 expression dropped to the level of uninjured striatum by 14 days after MCAO (revised Fig. 5).

Re-REVIEW: thank you for the inclusion of new data

Line 221 - what is the biological relevance of completing a phagocytosis assay at 4°C?

The incubation of cells at a low temperature is a commonly used negative control for ingestion. In such conditions, phagocytes can bind targets, but the internalization of particles is largely prevented (Peterson et al., Infect Immun 1977; Mondal et al., Comp Biochem Physiol A Mol Integr Physiol 2001; Mao et al., JCB 2009). It is thought that low temperatures reduce membrane fluidity and influence many other pertinent cellular properties or activities.

Line 241 - what are these wild type astrocytes? Earlier in the manuscript the authors state that their cell culture model already produces cells that are like stroke-induced reactive astrocytes.

Thank you for indicating this error. "Wild type astrocytes" has been replaced with "non-treated astrocytes" in the revised version of the manuscript (Line 262).

ABCA1 staining in Fig. 5a appears very high. Is there a baseline level of expression broadly across the cell surface of all astrocytes? - or is this simply non-specific staining of the antibody?

As you stated, ABCA1 signals seem to be expressed broadly throughout astrocytes. In addition, when astrocytes bind to beads, they are localized at the bead binding sites. To analyse their spatial distribution more precisely, we performed additional experiments, and the expression of ABCA1-GFP was forced in cultured astrocytes. ABCA1-GFP signals were localized broadly in the surface membrane and at the bead binding sites (arrowheads) as shown below. The distribution pattern of ABCA1-GFP was associated with ABCA1 as shown in Fig. 7a, and thus we think that ABCA1 signals are not simply non-specific staining of the antibody.

Re-REVIEW: A very interesting finding, and I thank the authors for this additional information.

Line 327 - do the transcript signatures of the pathophysiological astrocytes resemble developmental/embryonic/immature astrocytes? The literature does not support this comment.

As you stated, the original manuscript was speculative. We have toned down the expression as follows.

Re-REVIEW: Appreciated

Line 383

During brain development, network structures and functions dynamically change as well as after acute brain injury, and the brain becomes more plastic than normal adult brain under these situations. Accumulating evidence indicates both astrocytes and microglia play roles in remodelling during these dynamical plastic changes^{4, 56, 57, 58, 59, 60, 61, 62}. Therefore, there might be functional and genetic similarities, under pathophysiological conditions, adult astrocytes might transform into a more phagocytic state similar to immature and/or developmental astrocytes. Together, these findings suggest that astrocytes can phagocytize adjacent materials, including synapses, although this occurs primarily during development or under pathophysiological conditions in the adult brain.

Line 336 - which Dock?

Line 336 - which Elmo?

We apologize for the inadequate explanation. It is known that Crk II, Dock180 and Elmo1 mediate phagocytosis (Albert et al., Nat Cell Biol 2000; Gumienny et al., Cell 2001; Park et al., Nature 2007; Elliott et al., Nature 2010; Lu et al., Nat Cell Biol 2011). We have added these references and statements to the revised version of the manuscript (Line 407).

Re-REVIEW: Appreciated

Line 434 - two rectal thermometers?

We thank you for noticing the duplication. We have deleted the redundant sentence in the revised manuscript

.

Fig. 1 - state what the gold particles in immune-EM are labelling. Is it synaptophysin?

In Fig. 1, gold particles were used to label the GFAP antibody. Although we described this in Figure 1g and the legend in the original version of the manuscript, we have described this in the text body of the revised manuscript (Line 113).

Fig. 4 - continuity, place boxes around all or no panels.

We thank you for noticing this error. We have corrected this mistake in revised Fig. 6 (original Fig. 4).

In general protein is not 'expressed', mRNA is expressed. This should be fixed throughout the manuscript. Multiple continuity problems exist with gene and protein names. Ideally, genes should be italicised with a capitalised first letter (eg. *Gfap*, *Mertk*), while protein names should be capitalized (eg. GFAP, MERTK). This manuscript switches on several occasions - eg. *Mertk*, MERTK, MerTK.

We thank you for the comment regarding the incorrect terminology and abbreviations. We have deleted "expression" when describing proteins, and have corrected all gene and protein names in the revised version of the manuscript.

Re-REVIEW: Appreciated. The continuity makes understanding the complexity of the manuscript much easier for the reader.

There are several grammatical errors throughout the manuscript (eg. Line 50 '...thousands of debris...', line 196 'Although the another...') These should be carefully edited out of future versions of the manuscript.

We thank the reviewer for this comment. The revised manuscript has been edited and proofread by a native-English speaker.

REVIEWERS' COMMENTS in mail:

Reviewer #1 (Remarks to the Author):

Additional comments on new material:

Line 398 - how have the authors determined that the astrocytes they are studying in this study are 'A1' or 'A2'? There does not appear to be any sequencing data to validate this claim.

Re-Reply: We apologize that this part was confusing in the manuscript. We meant that "this study" is the original literature, i.e., "Neurotoxic reactive astrocytes are induced by activated microglia" written by Liddelow et al.,. In the re-revised manuscripts, we have now replaced "this study" with " this literature".

Line ~462 - Comments on spatiotemporal phagocytic capacity of astrocytes versus microglia is very much appreciated.

Overall, the manuscript has been much improved by these additions, and careful review of the manuscript combined with streamlining of the descriptions of the data contained. The conclusions now more appropriately match the data provided (in both the revised main figures, as well as the extensive supplementary material).